# CAN HYPERGRAPH MODELS BE STRONG BASELINES FOR NODE-LEVEL TASKS? SOLVING THE HYPEREDGE POLLUTION PROBLEM

## ABSTRACT

While homogeneous hypergraph-based algorithms have achieved promising performance on certain node-level tasks, they often exhibit sub-optimal results on others. This inconsistency is primarily attributed to structural flaws in hypergraph construction, with Hyperedge Pollution (HP) as a key factor. In this work, HPHNN is proposed to address the HP problem in hypergraph construction, improving the performance of classical hypergraph models and make them as strong baselines for node-level graph tasks. Experimental results on 11 real-world graph datasets demonstrate that hypergraph-based models not only outperform these baselines but also exhibit more stable and generalizable performance across diverse node-level tasks by mitigating the HP problem. Our findings challenge the prevailing view that GNNs and GTs inherently outperform other paradigms in node-level graph tasks, while offering a novel optimization perspective for the design of other hypergraph-based algorithms.

## 1 INTRODUCTION

Graph Neural Networks (GNNs) (Kipf & Welling, 2017; Velickovic et al., 2018; Zou et al., 2024; Duan et al., 2024a; Zhang et al., 2025a), and Graph Transformers (GTs)-based models (Yun et al., 2019; Wu et al., 2023; Deng et al., 2024; Xu et al., 2025; Luo et al., 2025) have achieved remarkable success in node-level graph learning ,task Wu et al. (2020), which concerns the attributes of nodes or label prediction. Recent advances in GNNs for node-level tasks include lightweight architectures for efficiency (He et al., 2020; Huang et al., 2023; Topping et al., 2022), masking-based reconstruction (Hou et al., 2022; Duan et al., 2024a), and contrastive learning (You et al., 2020; Wan et al., 2024) to mitigate overfitting and over-smoothing, and robust (Jin et al., 2020; Sun et al., 2020) designs to handle noisy graphs. Complementing these efforts, recent researches in GTs focus on improving scalability (Deng et al., 2024), incorporating positional and structural encodings (Black et al., 2024; Foumani et al., 2024). These elaborate designs, which focus on pairwise interactions, enhance the performance of both GNNs and GTs, yet struggle to capture the inherent relationships in the complex graph.

Hypergraph neural networks (HGNNs) (Feng et al., 2019; Yadati et al., 2019; Bai et al., 2021; Gao et al., 2022) have emerged as powerful tools for encoding group-wise semantics by modeling interactions among multiple nodes through hyperedges, offering better flexibility in representing complex structures. However, HGNNs with the capacity of modelling high-order information in graphs often lag behind GNNs and GTs in node-level tasks. A key reason lies in the **Hyperedge Pollution (HP)** problem (as illustrated in Fig. 1) introduced by a pre-defined hypergraph. When hyperedges are constructed manually or via similarity-based heuristics, nodes of distinct types in such a hypergraph may be erroneously assigned to the same hyperedge, resulting in high intra-hyperedge heterogeneity. According to the degree of heterogeneity, hyperedges can be categorized into: (a) high, (b) medium, and (c) low pollution. Lower pollution indicates a more homogeneous node composition, which is desirable for effective node-level representation learning.

Motivated by this, our work focuses on formally defining the HP problem and alleviating hyperedge pollution to improve the performance of hypergraph neural networks on node-level tasks. Specifically, we formally define the HP problem and characterize it with the designed HP

Figure 1: Illustration of hyperedge pollution. Nodes of different types may be incorrectly grouped into the same hyperedge, leading to varying levels of heterogeneity within hyperedges: (a) high, (b) medium, and (c) low pollution levels.

score of the hypergraph. Building upon this, we develop an HP-based dynamic hypergraph neural network (HPHNN) framework tailored for addressing the above-mentioned challenge of node-level tasks. The main strategy is to mitigate the pollution of the hypergraph by adaptively updating its structure, thereby improving hyperedge consistency. Specifically, the quality of the constructed hypergraph is evaluated by a designed dynamic hypergraph optimization module, using multiple criteria. These evaluation metrics facilitate the adaptive adjustment of hyperedge quantities, allowing for iterative and principled refinement of the hypergraph topology during training, thereby improving the robustness and expressiveness of the learned node embeddings. Extensive experiments on 11 benchmark datasets show that the proposed HPHNN method outperforms state-of-the-art GNN, GT, and hypergraph-based approaches on all datasets by effectively addressing the HP issue.

The contributions of this paper are summarized as follows:

- We identify the HP problem in hypergraph construction and propose the HP score as a metric to quantify it. The HP score captures the structural inconsistency within the hypergraph and offers clear insights into why hypergraph-based models may yield suboptimal results on specific node-level tasks.

- A HP-based dynamic hypergraph framework is proposed that iteratively refines the structure of the hypergraph through a dedicated dynamic hypergraph optimization module. This module leverages a quality score to guide the structural adjustment of the hypergraph, providing a promising direction for addressing the HP problem and enhancing hypergraph construction.

- Comprehensive experimental results show that the proposed method consistently outperforms a wide range of baseline approaches, providing empirical evidence that mitigating the HP problem leads to improved performance of hypergraphs in node-level tasks.

## 2 RELATED WORK

### 2.1 GRAPH NEURAL NETWORKS

Graph Neural Networks (GNNs) are fundamental to graph representation learning (Zhou et al., 2020; Xu et al., 2018), aggregating information from local neighbors via message passing. Kipf & Welling (2017) uses the normalized graph Laplacian matrix for convolution, while Velickovic et al. (2018)integrates attention mechanisms to assign weights to neighbors dynamically. To reduce reliance on labeled data, self-supervised methods such as GraphMAE (Hou et al., 2022) and GraphMVM (Duan et al., 2024a) train models to recover masked node features using the surrounding context. Dileo & Zignani (2024) finds that GNNs can surprisingly reconstruct node-level structural features like PageRank, centralities, and clustering coefficients from graph data. Zhang et al. (2025a) further improves performance under limited labels by decoupling embedding direction and norm, leveraging homophily among unlabeled nodes. Despite these advances, most GNNs depend on fixed and potentially noisy graph structures, limiting their expressiveness and robustness. To address this, Graph

Structure Learning (GSL) methods aim to optimize graph topology (Chen et al., 2019). For example, Duan et al. (2024b) builds encoding trees based on structural entropy to capture hierarchical communities, enhancing robustness and representation quality. However, most GNNs rely on pairwise message passing, which limits their ability to capture high-order interactions and complex node relationships. Additionally, their performance often suffers on noisy or suboptimal graph structures, highlighting the need for more flexible and expressive models like hypergraph networks or graph transformers that go beyond local pair-wise aggregation.

## 2.2 GRAPH TRANSFORMER NETWORKS

Graph Transformer Networks (GTs) extend Transformer architectures to graph-structured data (Ying et al., 2021) by capturing both local and global dependencies beyond traditional message-passing schemes. Early methods like GT(Yun et al., 2019) incorporate structural encodings into attention but struggle with complex graph topologies. Wu et al. (2023) enhances scalability by leveraging energy-constrained diffusion to capture long-range dependencies in large graphs. Francl et al. (2022) further enhances scalability through sparse attention. Chen et al. (2023) models implicit relations beyond observed edges using learnable positional encodings. Deng et al. (2024) adopts a polynomial norm-based attention for efficient global modeling, and Xu et al. (2025) leverages Laplacian information to integrate spectral features into graph attention. However, despite their success in modeling global interactions, existing graph transformers primarily focus on pairwise dependencies, limiting their ability to fully capture high-order relationships and complex semantic structures inherent in graphs. Moreover, their reliance on predefined or learned pairwise interactions can still be sensitive to noisy or incomplete graph data.

## 2.3 HYPERGRAPH NEURAL NETWORKS

Hypergraph Neural Networks have attracted growing attention due to their ability to model high-order relationships beyond pairwise edges in graphs. Static hypergraph methods construct a fixed hypergraph structure where each hyperedge connects multiple nodes, capturing complex group interactions. Early works such as HGNN and HyperGCN (Feng et al., 2019; Yadati et al., 2019) formulate the general Hypergraph Neural Network (HGNN) algorithm based on a two-stage message passing mechanism. A hypergraph is represented by an incidence matrix mapping nodes to hyperedges, and node features are updated through node-to-hyperedge and hyperedge-to-node aggregation. This framework enables HGNNs to capture higher-order group interactions that are inaccessible to traditional GNNs operating on pairwise edges. However, existing static hypergraph construction methods often produce noisy or heterogeneous hyperedges, which can harm node-level performance. Prior studies have shown that static constructions may introduce irrelevant or erroneous hyperedges (Cai et al., 2022), that ignoring node heterogeneity groups unrelated nodes into the same hyperedge and injects noise into message passing (Zhang et al., 2025b), and that heterogeneous hyperedges mix signals from different classes and weaken decision boundaries (Yin et al., 2025). These findings collectively indicate that hyperedge heterogeneity is a key factor affecting HGNN performance. To address this, hypergraph structure learning methods have been proposed to dynamically refine hypergraph connections. Dynamic hypergraph neural networks (DHGNNs) aim to uncover latent connections from node features, enabling more informative hypergraph representations. For example, Jiang et al. (2019) reconstructed hypergraphs by integrating KNN and K-means to capture both local proximity and global structure. Bai et al. (2021)introduced a continuous incidence matrix with attention-based connection weights to enhance structural flexibility. Zhou et al. (2023)dynamically adjusts the number of hyperedges based on the saturation level of the constructed hyperedges. Although existing methods like DHGNN and TDHNN dynamically construct hypergraphs, they lack explicit evaluation of hyperedge quality. Most approaches assume all constructed hyperedges are informative, which can introduce noise. In contrast, our method assesses hyperedge pollution after construction and uses it to guide the next round, enabling more accurate and robust hypergraph modeling.

## 3 METHOD

Fig. 2 illustrates the overall architecture of our proposed method, which will be detailed in the following section.

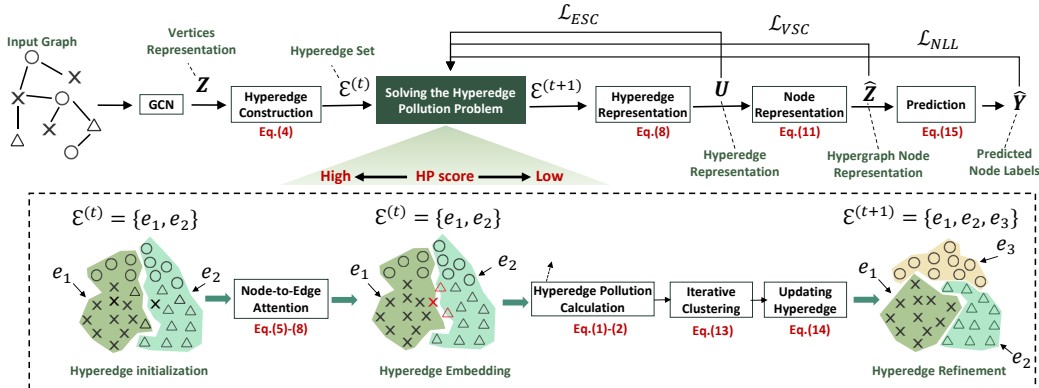

Figure 2: The overall framework of the proposed model.

## 3.1 PROBLEM FORMULATION

Formally, a hypergraph is defined as $\mathcal{G} = \{V, \mathcal{E}, \mathbf{X}, \mathbf{U}\}$, where $V = \{v_1, v_2, \ldots, v_n\}$ and $\mathcal{E} = \{e_1, e_2, \ldots, e_m\}$ denotes the set of $n$ vertices and $m$ hyperedges, respectively. The matrix $\mathbf{X} \in \mathbb{R}^{n \times d_v}$ encodes $d_v$-dimensional features for each vertex, while the matrix $\mathbf{U} \in \mathbb{R}^{m \times d_e}$ represents $d_e$-dimensional features for hyperedge. Unlike the original graph, the hyperedge includes more than two vertices. Thus, the structure of a hypergraph is captured by an incidence matrix $\mathbf{H} \in \mathbb{R}^{n \times m}$, where each entry $h_{i,j}$ indicates the relationship between node $v_i$ and hyperedge $e_j$. Specifically, $\mathbf{H}_{i,j} = 1$ denotes that hyperedge $e_j$ contains the node $v_i$; otherwise, $\mathbf{H}_{i,j} = 0$.

As discussed in Section 1, certain hypergraph-based models exhibit suboptimal performance on node-level tasks. A primary issue stems from the inconsistency among vertices within the same hyperedge, termed the hyperedge pollution (HP) problem. We provide a formal definition of the pollution score to quantify this inconsistency.

**Definition 1.** Supposing that the hypergraph $\mathcal{G}$ contains $\mathcal{E}$ hyperedges and the hyperedge $e_k$ has $|e_k|$ vertices, the pollution score of $e_k$ is formulated as

$$\text{HP}(e_k) = 1 - \frac{1}{|e_k|^2} \sum_{i,j \in e_k} \left( \frac{\mathbf{X}_{i,:} \cdot \mathbf{X}_{j,:}}{||\mathbf{X}_{i,:}|| \cdot ||\mathbf{X}_{j,:}||} \right). \tag{1}$$

Then, the pollution score of the hypergraph is computed by

$$\mathcal{HP}(\mathcal{E}) = \frac{1}{|\mathcal{E}|} \sum_{e_k \in \mathcal{E}} \text{HP}(e_k). \tag{2}$$

Based on the definition, the pollution score serves as an indicator of intra-hyperedge structural coherence. If all nodes within the hyperedge $e_k$ are highly similar, the cosine similarity $\frac{\mathbf{X}_{i,:} \cdot \mathbf{X}_{j,:}}{||\mathbf{X}_{i,:}|| \cdot ||\mathbf{X}_{j,:}||}$ approaches 1, leading $\text{HP}(e_k)$ tend to be 0. In contrast, a large pollution score $\text{HP}(e_k)$ indicates a decreased homogeneity within the hyperedge, caused by the presence of multiple node types. Likewise, the pollution score of a hypergraph qualifies the global structural consistency by averaging the pollution scores of all hyperedges. A lower $\mathcal{HP}(\mathcal{E})$ indicates higher node similarity coherence in hyperedges, while a higher value reflects greater structural inconsistency. The detailed theoretical analysis is shown in Appendix A.

In summary, the definition offers two complementary perspectives for assessing structural irregularity: the local pollution score characterizes inconsistency within individual hyperedges, whereas the global variant captures the overall topological coherence of the hypergraph as a whole.

## 3.2 THE PROPOSED HPHNN

Building upon the identified HP issue of hypergraph, we introduce HPHNN, an algorithm designed to iteratively refine hypergraph structures and reduce pollution scores, thereby achieving strong performance in node-level tasks. The details of HPHNN are described below:

### 3.2.1 Hyperedge Embedding with Feature Aggregation

To obtain a robust initialization, the features of the vertices are fed into a two-layer GNN, which can learn the representations of all vertices by integrating the information of neighbors. Next, an original graph $G$ is derived using the similarity measurement, with the corresponding similarity matrix denoted $\mathbf{A} \in \mathbb{R}^{n \times n}$. The new vertice representation $\mathbf{Z}$ is computed by

$$\mathbf{Z} = \sigma\left(\mathbf{A} \cdot \sigma\left(\mathbf{A} \cdot \mathbf{XW}\right) \cdot \mathbf{W}'\right), \quad \mathbf{A} = \mathbf{D}^{-\frac{1}{2}}(\mathbf{A} + \mathbf{I})\mathbf{D}^{-\frac{1}{2}}, \quad \mathbf{D}_{ii} = \sum_j \mathbf{A}_{i,j}, \tag{3}$$

where $\mathbf{D}$ is the diagonal degree matrix of the graph $G$ and $\mathbf{I}$ is the identity matrix.

Subsequently, HPHNN constructs the hypergraph based on the learned vertex representations. Specifically, the $k$-means algorithm is applied to partition the vertices into $m$ clusters, where the number of clusters corresponds to the number of initial hyperedges. Thus, we have

$$\mathcal{E} = \texttt{Clustering}(\mathbf{Z}, m) = [e_1, e_2, \dots, e_m]. \tag{4}$$

Following that, each hyperedge is represented by an embedding computed through the aggregation of the features of the vertices within the corresponding cluster. The initial embeddings of the hyperedges are formulated as follows:

$$\mathbf{U} = [\mathbf{u}_1, \mathbf{u}_2, \dots, \mathbf{u}_m], \quad \mathbf{u}_k = \frac{1}{|e_k|} \sum_{i \in e_k} \mathbf{Z}_{i,:} \quad \text{for} \quad k = 1, 2, \dots, m. \tag{5}$$

### 3.2.2 Hyperedge Updating with Attention Mechanism

While the initial embeddings provide a rough approximation of hyperedge semantics, they do not account for fine-grained dependencies among vertices. To enhance expressiveness, we apply an attention mechanism to update the hyperedge features during training iteratively.

$$\mathbf{Q} = \mathbf{UW}^Q, \quad \mathbf{K} = \mathbf{ZW}^K, \quad \alpha_{e_i, v_j} = \frac{\exp\left(\mathbf{Q}_{i,:} \cdot \mathbf{K}_{j,:}^\top\right)}{\sum_{l=1}^n \exp\left(\mathbf{Q}_{i,:} \cdot \mathbf{K}_{l,:}^\top\right)}. \tag{6}$$

The attention coefficient $\alpha_{e_i, v_j}$ represents the contribution of vertex $v_j$ to hyperedge $e_i$. For each hyperedge, we select the top-$k_n$ vertices according to their attention coefficients and denote them as

$$\mathcal{N}_i = \left\{ j \,\middle|\, j \in \text{top} - k_n\left(\left\{\alpha_{e_i, v_j}\right\}_{j=1}^n\right) \right\}. \tag{7}$$

After normalizing the attention coefficients, we update the hyperedge embedding with the weighted representations of its top-$k_n$ related vertices. The detailed computation is shown as follows:

$$\tilde{\alpha}_{e_i, v_j} = \frac{\alpha_{e_i, v_j}}{\sum_{k \in \mathcal{N}_i} \alpha_{e_i, v_k}}, \quad \mathbf{u}_k = \sum_{j \in \mathcal{N}_i} \tilde{\alpha}_{e_i, v_j} \mathbf{X}_{j,:} \mathbf{W}^V. \tag{8}$$

### 3.2.3 Hypergraph Structure Refinement with Top-$k_e$ Hyperedges

Although the fused feature of the hyperedge effectively captures the essential information from its $k_n$ most similar vertices, the set of vertices associated with the hyperedge is decided by the clustering algorithm. The clustering-based assignment of vertices may associate a node with a hyperedge that is not semantically relevant due to noise or limited contextual information. Instead of relying on the initial assignments, we refine the structure of the hypergraph with the updated hyperedge embedding. The computation is described below:

$$\widehat{\mathbf{Q}} = \mathbf{Z}\widehat{\mathbf{W}}^Q, \quad \widehat{\mathbf{K}} = \mathbf{U}\widehat{\mathbf{W}}^K, \quad \alpha_{v_i, e_j} = \frac{\exp\left(\widehat{\mathbf{Q}}_{i,:} \cdot \widehat{\mathbf{K}}_{j,:}^\top\right)}{\sum_{l=1}^m \exp\left(\widehat{\mathbf{Q}}_{i,:} \cdot \widehat{\mathbf{K}}_{l,:}^\top\right)} \tag{9}$$

By selecting only the top-$k_e$ hyperedges for each node based on attention scores, we construct a sparse and semantically meaningful incidence matrix $\mathbf{H}$, which helps reduce noise and improve the representation quality of the learned hypergraph structure. Thus, the incidence matrix is updated by

$$\widehat{\mathcal{N}}_i = \left\{ j \,\middle|\, j \in \text{top} - k_e\left(\left\{\alpha_{v_i, e_j}\right\}_{j=1}^m\right) \right\}, \quad \mathbf{H}_{i,j} = \begin{cases} \alpha_{v_i, e_j}, & \text{if } j \in \widehat{\mathcal{N}}_i, \quad i = 1, 2, \dots, n \\ 0, & \text{otherwise} \end{cases}. \tag{10}$$

Accordingly, the representations of vertices are also updated with the hyperedge representation and attention coefficients. The specific process is as follows:

$$\widehat{\mathbf{Z}} = \mathbf{Z}\mathbf{W}_1 + \mathbf{H}\mathbf{U}\mathbf{W}_2, \tag{11}$$

In summary, this equation allows the model to update vertex representations by combining both local features (from the vertex itself) and global features (from the hypergraph structure), making it more effective for node classification or other graph-based learning tasks.

### 3.2.4 ADJUSTMENT OF HYPEREDGE NUMBER

Although the vertices are divided into different hyperedges based on the similarity measurement, the hypergraph with a fixed number of hyperedges is unsatiable for capturing the inherent information on all datasets. Inspired by the dynamic hypergraph model, the structure of the learned hypergraph will be updated by using a designed hypergraph quality score $\mathcal{S}_{\mathcal{H}}$, which is

$$\mathcal{S}_{\mathcal{H}} = \text{Satur}(\mathcal{E}) + \text{Silh}(\widehat{\mathbf{Z}}, \mathbf{Y}) - \mathcal{HP}(\mathcal{E}), \tag{12}$$

where $\mathbf{Y}$ is the label of vertices. The quality score has three terms to ensure the robustness and accuracy of the learned hypergraph. The first term $\text{Satur}(\mathcal{E})$ refers to the saturation of the hyperedges, which represents the density of the involvement of vertices in hyperedges. In other words, it measures the number of vertices participating in the hyperedges. The second term $\text{Silh}(\widehat{\mathbf{Z}}, \mathbf{Y})$ represents the silhouette score of the vertices $\widehat{\mathbf{Z}}$ and labels $\mathbf{Y}$, which aims to keep the quality of classification. The third term $\mathcal{HP}(\mathcal{E})$ represents the pollution of a hypergraph mentioned in definition 2, which quantifies the consistency of the hyperedges.

The score contains three terms that jointly ensure the stability and reliability of the learned hypergraph.

**(1) Hyperedge saturation.** $\text{Satur}(\mathcal{E})$ measures the proportion of non-empty hyperedges and is introduced in TDHNN(Zhou et al., 2023). It is formally defined as $\text{Satur}(\mathcal{E}) = 1 - \frac{|E_{\text{empty}}|}{|E|}$, where $E_{\text{empty}}$ denotes the set of empty hyperedges. This term discourages redundant or empty hyperedges during hypergraph reconstruction and promotes more stable structural updates.

**(2) Silhouette score.** $\text{Silh}(\widehat{\mathbf{Z}}, \mathbf{Y})$ is the standard silhouette coefficient from clustering analysis(Rousseeuw, 1987), quantifying how well vertex embeddings $\widehat{\mathbf{Z}}$ align with the class labels $\mathbf{Y}$. A higher score indicates better intra-class compactness and inter-class separability. Including this term encourages the updated hypergraph to preserve discriminative patterns relevant to node classification.

**(3) Hyperedge pollution.** $\mathcal{HP}(\mathcal{E})$ measures the semantic inconsistency within hyperedges (Definition 2). A lower pollution score indicates more coherent hyperedges. Subtracting this term penalizes hyperedge heterogeneity and encourages structurally meaningful updates.

Together, these three metrics guide the hypergraph toward a balanced structure that avoids empty hyperedges, preserves discriminative vertex relations, and reduces semantic inconsistency.

A hypergraph with coherent vertices within hyperedges typically exhibits high saturation and silhouette scores but low pollution, leading to a high overall quality score. However, the higher quality score does not necessarily indicate a better structure in the hypergraph. Focusing too much on quality score may overly optimize the pollution score $\mathcal{HP}(\mathcal{E})$ while overlooking the diversity between vertices. Thus, we set a threshold to control this situation. If the threshold is larger than the value $\theta$, we add a hyperedge to the hypergraph. If the threshold is smaller than the value $\theta$, we delete a hyperedge from the hypergraph. Specifically, the number of hyperedges of layer $t + 1$ is decided by the quality score of the hypergraph at layer $t$, that is

$$m^{(t+1)} = \begin{cases} m^{(t)} + 1, & \text{if } S_{\mathcal{H}}^{(t)} > \theta \\ m^{(t)} - 1, & \text{if } S_{\mathcal{H}}^{(t)} < \theta \end{cases}. \tag{13}$$

When the number of hyperedges is updated to $m^{(t+1)}$, we classify all vertices into $m^{(t+1)}$ clusters with k-means algorithms. So we have

$$\mathcal{E}^{(t+1)} = \texttt{Clustering}(\widehat{\mathbf{Z}}^{(t)}, m^{(t+1)}) = [e_1, e_2, \ldots, e_{m^{(t+1)}}] \tag{14}$$

### 3.2.5 Loss Function of the Proposed HPHNN

At last, the representations of all vertices are fed to the MLP and normalized to obtain the predicted label $\widehat{\mathbf{Y}}$. The detailed computation is

$$\widehat{\mathbf{Y}} = [\widehat{\mathbf{y}}_1, \widehat{\mathbf{y}}_2, \ldots, \widehat{\mathbf{y}}_n], \quad \widehat{\mathbf{y}}_i = \log\left(\frac{\exp\left(\text{MLP}(\widehat{\mathbf{Z}}_i)\right)}{\sum_{j=1}^{n} \exp\left(\text{MLP}(\widehat{\mathbf{Z}}_j)\right)}\right). \tag{15}$$

The loss function also has three parts, that is

$$\mathcal{L} = \mathcal{L}_{NLL} + \mathcal{L}_{VSC} + \mathcal{L}_{\text{ESC}}. \tag{16}$$

The $\mathcal{L}_{NLL}$ is the negative log-likelihood loss, which is a commonly used loss function in classification. The equation is

$$\mathcal{L}_{NLL} = -\frac{1}{|\mathbf{n}|} \sum_{i \in \mathbf{n}} \mathbf{y}_i^\top \log \widehat{\mathbf{y}}_i. \tag{17}$$

It measures the difference between the predicted label and the ground truth. The $\mathcal{L}_{VSC}$ refers to the vertices similarity constraint loss. We compute it by

$$\mathcal{L}_{VSC} = \sum_{e_k \in \mathcal{E}} \sum_{i=1}^{|e_k|} \left\|\widehat{\mathbf{Z}}_{i,:} - \widehat{\mathbf{Z}}_{\pi(i),:}\right\|_2^2, \tag{18}$$

where $\pi(i)$ is the $i$-th vertex of the random arrangement of the vertices on the hyperedge $e_k$. It measures the distance among the vertices of the hypergraph, leading to local consistency. The $\mathcal{L}_{ESC}$ represents the embedding structure consistency loss of hyperedges. The formulation is

$$\mathcal{L}_{\text{ESC}} = -\frac{1}{m} \sum_{i=1}^{m} \left\|\mathbf{U}_i - \mathbf{U}_{\pi(i)}\right\|^2. \tag{19}$$

It measures the dissimilarity among the embeddings of the hyperedges, while keeping the diversity of the hyperedges.

## 4 Experiments

### 4.1 Datasets and Baselines

We evaluate HPHNN on eleven benchmarks: three citation networks – Cora, Citeseer and Pubmed (Kipf & Welling, 2017); one social graph Twitch-PT; two 3D object datasets ModelNet40 (Wu et al., 2015) and NTU2012 (Chen et al., 2003); two biomedical networks Disease and Leukemia (Li et al., 2023); one Airport transportation network (Chami et al., 2019); and two e-commerce graphs Computers (McAuley et al., 2015) and Products (Hu et al., 2020). Dataset statistics are in Appendix C.

A total of 18 baseline methods are selected as comparative models, categorized into three groups: **Graph neural network-based**: GCN, GAT, UniG-Encoder, GraphMVM, WGNN, and Norm-Prop(Kipf & Welling, 2017; Velickovic et al., 2018; Zou et al., 2024; Duan et al., 2024a; Ji et al., 2023; Zhang et al., 2025a); **Graph Transformer-based**: GT, Exphoermer, DIFFormer, NAG-phormer, Polynormer, and FairGP(Yun et al., 2019; Francl et al., 2022; Wu et al., 2023; Chen et al., 2023; Deng et al., 2024; Luo et al., 2025); **Hypergraph-based**: HGNN, HyperGCN, HyperConv, DHGNN, TDHNN, and DHHNN(Feng et al., 2019; Yadati et al., 2019; Bai et al., 2021; Jiang et al., 2019; Zhou et al., 2023; Mei et al., 2025).

Table 1: Acc% Comparison Results on Eleven Real-world Datasets. Δ denotes the improvement of the best result over the second-best.

| Method | Computers | Cora | Citeseer | NTU2012 | Products | ModelNet40 | Pubmed | Airport | Twitch-PT | Disease | Leukemia |
|---|---|---|---|---|---|---|---|---|---|---|---|
| GCN | 86.95 ± 0.52 | 86.98 ± 1.27 | 76.50 ± 1.36 | 80.43 ± 1.09 | 75.64 ± 0.21 | 94.85 ± 1.75 | 87.42 ± 0.37 | 55.60 ± 1.20 | 68.90 ± 0.60 | 55.30 ± 0.90 | 58.20 ± 0.10 |
| GAT | 90.78 ± 0.13 | 87.30 ± 1.10 | 76.55 ± 1.23 | 80.16 ± 1.08 | 79.45 ± 0.59 | 95.75 ± 0.14 | 86.33 ± 1.48 | 67.00 ± 0.80 | 66.40 ± 0.70 | 56.05 ± 0.83 | 58.70 ± 0.50 |
| UniG-Encoder | 90.38 ± 0.47 | 87.36 ± 1.17 | 77.33 ± 1.86 | 92.43 ± 1.49 | 81.35 ± 0.72 | 94.36 ± 0.86 | 89.76 ± 0.37 | 95.77 ± 1.32 | 65.62 ± 0.50 | 93.68 ± 1.43 | 62.14 ± 0.63 |
| WGNN(2023) | 91.83 ± 0.45 | 90.04 ± 0.90 | 77.68 ± 0.93 | 91.69 ± 1.26 | 81.73 ± 0.47 | 93.58 ± 1.02 | 88.69 ± 0.38 | 93.18 ± 1.22 | 69.98 ± 1.40 | 94.56 ± 0.53 | 60.61 ± 0.47 |
| GraphMVM(2024) | 92.82 ± 0.24 | 89.25 ± 0.37 | 78.39 ± 0.55 | 91.85 ± 1.69 | OOM | 94.69 ± 0.85 | 88.58 ± 0.29 | 94.52 ± 1.73 | 69.36 ± 1.01 | 91.14 ± 1.76 | 60.79 ± 1.04 |
| NormProp(2025) | 92.75 ± 0.58 | 89.48 ± 0.56 | 80.59 ± 0.57 | 92.21 ± 0.86 | 82.57 ± 0.42 | 95.58 ± 1.45 | 88.95 ± 1.35 | 96.89 ± 1.48 | 72.01 ± 0.37 | 94.38 ± 0.94 | 61.12 ± 0.10 |
| GT | 91.18 ± 0.17 | 87.49 ± 0.34 | 78.12 ± 0.42 | 91.12 ± 1.04 | OOM | 92.58 ± 0.76 | 88.49 ± 0.13 | 95.38 ± 0.63 | 67.68 ± 0.46 | 92.76 ± 0.83 | 59.62 ± 0.26 |
| Exphoermer(2023) | 91.59 ± 0.31 | 88.96 ± 0.26 | 79.65 ± 0.24 | 91.29 ± 1.27 | OOM | 93.91 ± 0.59 | 89.12 ± 0.19 | 96.24 ± 0.58 | 70.52 ± 0.27 | 93.28 ± 0.61 | 60.18 ± 0.26 |
| DIFFormer(2023) | 91.99 ± 0.76 | 89.04 ± 1.23 | 78.47 ± 1.16 | 90.73 ± 0.84 | 74.16 ± 0.31 | 94.21 ± 0.79 | 88.67 ± 0.50 | 96.92 ± 0.89 | 69.45 ± 0.21 | 91.91 ± 1.26 | 60.84 ± 1.23 |
| NAGphormer(2023) | 91.22 ± 0.14 | 88.30 ± 0.24 | 80.57 ± 0.72 | 91.24 ± 0.68 | 73.55 ± 0.21 | 94.62 ± 0.59 | 89.65 ± 0.24 | 98.72 ± 0.27 | 69.17 ± 0.84 | 94.94 ± 0.34 | 63.09 ± 0.80 |
| Polynormer(2024) | 93.68 ± 0.21 | 89.45 ± 1.37 | 79.79 ± 0.54 | 91.25 ± 1.58 | 83.82 ± 0.11 | 95.63 ± 0.86 | 90.10 ± 0.22 | 96.38 ± 1.68 | 71.02 ± 0.44 | 93.89 ± 0.63 | 62.45 ± 0.91 |
| FairGP(2025) | 91.72 ± 0.86 | 90.11 ± 0.59 | 82.11 ± 0.86 | 92.68 ± 1.36 | 82.79 ± 0.34 | 96.72 ± 1.45 | 89.39 ± 0.35 | 95.64 ± 0.35 | 69.88 ± 0.14 | 94.65 ± 0.78 | 61.61 ± 0.05 |
| HGNN | 87.30 ± 1.50 | 79.39 ± 1.36 | 72.45 ± 1.16 | 83.64 ± 0.37 | 78.70 ± 0.96 | 96.96 ± 1.43 | 86.44 ± 0.44 | 93.62 ± 0.57 | 69.50 ± 0.40 | 87.80 ± 1.83 | 58.67 ± 0.92 |
| HyperGCN | 88.10 ± 1.60 | 78.45 ± 1.26 | 71.28 ± 0.82 | 79.90 ± 0.91 | 79.76 ± 0.75 | 96.10 ± 0.63 | 82.84 ± 8.67 | 94.63 ± 0.37 | 69.80 ± 0.35 | 88.75 ± 1.89 | 59.01 ± 1.02 |
| HyperConv | 88.50 ± 1.78 | 86.93 ± 0.64 | 76.89 ± 0.45 | 84.69 ± 0.20 | 81.49 ± 0.86 | 95.74 ± 0.41 | 85.67 ± 0.52 | 93.42 ± 0.50 | 70.10 ± 0.50 | 89.79 ± 0.10 | 59.30 ± 0.80 |
| HAIN | 88.42 ± 1.23 | 82.31 ± 1.19 | 71.54 ± 1.22 | 85.17 ± 1.18 | 79.03 ± 1.15 | 95.12 ± 1.13 | 87.04 ± 1.21 | 89.02 ± 1.14 | 69.10 ± 1.20 | 89.32 ± 1.16 | 59.51 ± 1.25 |
| DHGNN | 89.20 ± 1.40 | 79.52 ± 1.19 | 73.59 ± 1.65 | 85.31 ± 0.26 | 80.46 ± 0.83 | 96.99 ± 1.46 | 84.24 ± 1.35 | 93.39 ± 0.82 | 71.00 ± 0.40 | 90.74 ± 0.57 | 60.25 ± 0.76 |
| TDHNN(2023) | 92.60 ± 0.24 | 88.67 ± 0.53 | 79.57 ± 0.24 | 86.05 ± 1.10 | 82.97 ± 0.64 | 97.52 ± 0.80 | 87.48 ± 0.79 | 98.43 ± 0.14 | 72.86 ± 0.41 | 94.94 ± 0.47 | 61.18 ± 1.66 |
| DHHNN(2025) | 92.01 ± 0.45 | 89.29 ± 0.62 | 80.92 ± 0.13 | 92.35 ± 0.59 | 84.29 ± 0.58 | 96.89 ± 0.76 | 89.47 ± 0.29 | 97.70 ± 0.56 | 69.61 ± 0.00 | 95.30 ± 0.69 | 61.26 ± 0.47 |
| **HPHNN** | 96.82 ± 0.13 | 92.85 ± 0.14 | 84.55 ± 0.39 | 94.52 ± 0.26 | 85.69 ± 0.67 | 98.75 ± 0.31 | 91.21 ± 0.12 | 99.79 ± 0.15 | 73.79 ± 0.24 | 96.05 ± 0.31 | 63.67 ± 0.09 |
| Δ | ↑ 3.14 | ↑ 2.74 | ↑ 2.44 | ↑ 1.84 | ↑ 1.40 | ↑ 1.23 | ↑ 1.11 | ↑ 1.07 | ↑ 0.93 | ↑ 0.75 | ↑ 0.58 |

## 4.2 RESULTS ANALYSIS

To evaluate the effectiveness of HPHNN, we conduct comprehensive experiments on eleven real-world datasets. For clarity, we highlight the best-performing results in red and the second-best in green throughout the tables. As shown in Table 1, our method consistently achieves the highest accuracy across all datasets, outperforming strong baselines and recent hypergraph-specific models such as DHGNN and DHHNN. Besides, we have the following observations:

Firstly, the high-order information encoded in hypergraphs has been shown to improve node classification performance significantly. As reported, most hypergraph-based methods consistently outperform graph-based and transformer-based models. For instance, on the ModelNet40 dataset, Norm-Prop and FairGP achieve 95.58% and 96.72% accuracy, respectively, while TDHNN improves the results to 97.52%, illustrating the advantage of modelling high-order relationships among nodes. Secondly, static hypergraph models suffer from the situation where irrelevant nodes are grouped. In contrast, dynamic hypergraph methods alleviate this issue by adaptively refining hyperedges, thereby improving performance. For example, DHHNN achieves 95.30%, outperforming all static hypergraph models on the Disease dataset. Our proposed method, HPHNN, further advances this line of work, achieving the highest accuracy of 96.05%. This superior performance stems from two key designs: (1) we introduce a pollution score to assess hyperedge quality and regulate hyperedge generation accordingly, and (2) we employ an attention-based node selection mechanism to filter noisy or irrelevant nodes within each hyperedge dynamically. Together, these mechanisms enable HPHNN to construct cleaner, semantically meaningful hyperedges, thus achieving state-of-the-art results with improved robustness.

To highlight the superiority of HPHNN in addressing the HP problem, we compare its performance with other hypergraph-based models by evaluating the pollution score. As shown in Fig. 3(a), unsurprisingly, original hypergraphs with fixed structure have the highest pollution scores, followed by dynamic updating ones. As expected, the HPHNN achieves the lowest pollution scores across all datasets, outperforming other hypergraph-based models with scores as low as 0.15. This is due to its dynamic hypergraph refinement, which helps avoid the HP problem while enhancing node representation learning. Furthermore, we analyze the changes in HP scores before and after training of HPHNN to evaluate its ability to mitigate semantic inconsistency within hyperedges. As shown in Fig. 3(b, c). Specifically, we selected five representative hyperedges (2 high, 1 medium, and 2 low) before and after training based on pollution scores. It is obvious that HPHNN significantly reduces hyperedge pollution after training, indicating enhanced semantic consistency among grouped nodes.

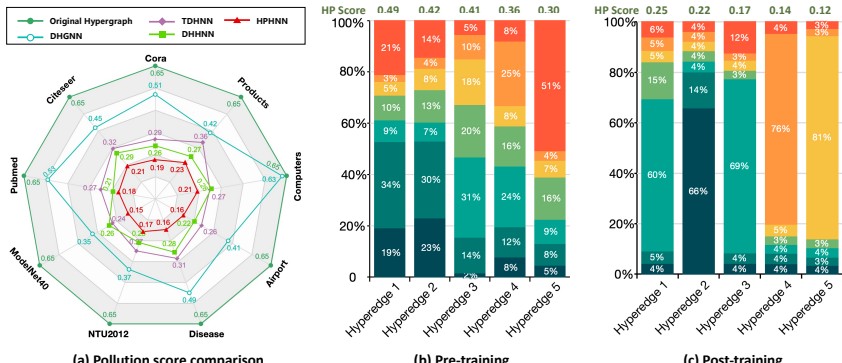

Figure 3: (a) Radar chart of hyperedge pollution (HP) score across nine datasets under different hypergraph construction methods. Category proportion statistics within hyperedges of the hypergraph constructed by HPHNN before and after training (b and c).

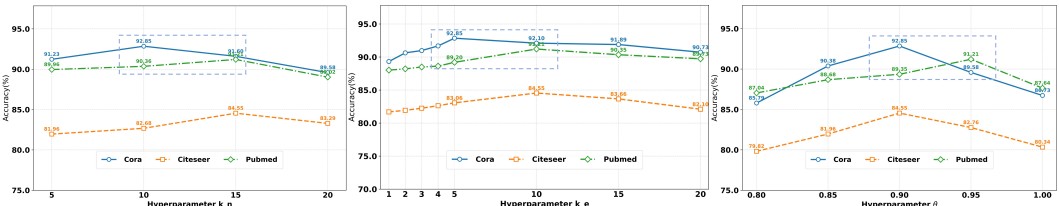

Figure 4: Analysis of three hyperparameters: $k_n$, $k_e$, and $\theta$. Results are shown on the Cora, Citeseer, and PubMed datasets.

### 4.3 ABLATION STUDY

To evaluate the contribution of two essential components in our model, we conduct an ablation study on eight datasets by selectively removing the two proposed components: hypergraph optimization via pollution score (HPS) in Eq. (12), and the hyperedge selection and refinement (HSR) in Eq. (10). As shown in Table 2, removing HPS, which dynamically adjusts hyperedge construction based on node-level noise, results in an average accuracy drop of 1.95%, confirming its ability to suppress noise-induced hyperedges. Excluding HSR, which filters out irrelevant hyperedges and propagates only the most similar ones to each node, also harms performance with an average drop of 1.27%, highlighting the benefit of selective hyperedge-to-node message passing. When all modules are combined, the model achieves the best results across all datasets, demonstrating that the two components are complementary and jointly enhance both accuracy and robustness.

### 4.4 PARAMETER AND COMPLEXITY ANALYSIS

To evaluate the robustness of our model, we analyze the impact of three key hyperparameters: the number of nearest neighbors for vertices $k_n$ and hyperedges $k_e$, and the quality score threshold $\theta$. Results on Cora, Citeseer, and PubMed are shown in Fig. 4. We find that the model performs consistently well when $k_n \in [10, 15]$, suggesting that a moderate number of intra-hyperedge neighbors promotes semantic consistency while mitigating over-smoothing. Likewise, performance remains stable for $k_e \in [5, 10]$, indicating that a small set of representative hyperedges suffices to capture high-order context. For the quality threshold $\theta$, which dynamically adjusts the number of hyperedges based on structural quality, optimal results are achieved in the $[0.90, 0.95]$. This setting strikes a balance between reducing hyperedge pollution and preserving classification accuracy. Overall, the model exhibits strong performance across a wide hyperparameter range, demonstrating its robustness and low sensitivity to parameter tuning. We also provide a detailed analysis of training time and GPU memory usage in Appendix B.

Table 2: Ablation Study of Each Module (HPS, HSR) on Eight Datasets with Accuracy (%).

| Module | Cora | Citeseer | Pubmed | Computers | ModelNet40 | Airport | Disease | Products |
|---|---|---|---|---|---|---|---|---|
| HPS Only | 91.93 | 83.24 | 89.68 | 95.49 | 97.06 | 98.64 | 94.82 | 84.68 |
| HSR Only | 90.19 | 81.98 | 89.67 | 94.79 | 96.83 | 97.98 | 95.63 | 83.01 |
| HPS + HSR | **92.85** | **84.55** | **91.21** | **96.82** | **98.75** | **99.79** | **96.05** | **85.69** |

## 5 CONCLUSION AND FUTURE WORK

In this work, we formally defined the problem of hyperedge pollution and identified hyperedge pollution as a key factor affecting semantic consistency in hypergraphs, explaining why existing hypergraph-based methods may underperform on specific node-level tasks. To address this issue, we proposed HPHNN, which introduced the pollution score to quantify the degree of heterogeneity within hyperedges and further incorporated it into a unified quality score that guides the dynamic refinement of the hypergraph structure. The proposed HPHNN achieves consistently strong performance across multiple benchmark datasets while effectively reducing hyperedge pollution. These results offer new insights for the design of hypergraph models, especially in terms of enhancing classification performance by mitigating structural inconsistency.

Despite these promising results, our approach still has limitations. Specifically, it requires sufficient training data for stable performance and incurs relatively high computational cost due to repeated clustering and hypergraph updates. In future work, we aim to address these issues by exploring semi- and self-supervised strategies to reduce reliance on labeled data. We also plan to design more fine-grained, node-level dynamic hypergraph construction methods and optimize time complexity.

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

## A    THEORETICAL ANALYSIS OF POLLUTION SCORE

### A.1    PRELIMINARIES

We recall the definition of the pollution score from Definition 1. Given a hypergraph $\mathcal{G}$ contains $\mathcal{E}$ hyperedges and the hyperedge $e_k$ has $|e_k|$ vertices, and its vertices feature is $\mathbf{X} \in \mathbb{R}^{n \times d}$, the pollution score is defined as:

$$\text{HP}(e_k) = 1 - \frac{1}{|e_k|^2} \sum_{i,j \in e_k} \left( \frac{\mathbf{X}_{i,:} \cdot \mathbf{X}_{j,:}}{||\mathbf{X}_{i,:}|| \cdot ||\mathbf{X}_{j,:}||} \right). \tag{20}$$

The global pollution score of the hypergraph is the average over all hyperedges:

$$\mathcal{HP}(\mathcal{E}) = \frac{1}{|\mathcal{E}|} \sum_{e_k \in \mathcal{E}} \text{HP}(e_k) \tag{21}$$

### A.2    KEY PROPERTIES

Based on the definition, we analyze the lower and upper bounds of $\text{HP}(e_k)$ and $\mathcal{HP}(\mathcal{E})$.

**Theorem 1.** *(Lower bound) The pollution score $HP(e_k)$ and $\mathcal{HP}(\mathcal{E})$ are both lower-bounded by 0.*

*Proof.* The cosine similarity $\cos(\theta_{ij}) = \frac{\mathbf{X}_i \cdot \mathbf{X}_j}{||\mathbf{X}_i|| \cdot ||\mathbf{X}_j||} \in [-1, 1]$. In practice, node embeddings are typically non-negative and normalized, resulting in cosine similarity values lying within the range $[0, 1]$. Supposing that all vertices within the hyperedge $e_k$ have the same features, we have

$$\begin{aligned} \text{HP}(e_k) &= 1 - \frac{1}{|e_k|^2} \sum_{i \in e_k} \left( \sum_{j \in e_k} \frac{\mathbf{X}_{i,:} \cdot \mathbf{X}_{j,:}}{||\mathbf{X}_{i,:}|| \cdot ||\mathbf{X}_{j,:}||} \right) \\ &= 1 - \frac{1}{|e_k|^2} \cdot |e_k| \cdot (|e_k| \cdot 1) \\ &= 0. \end{aligned} \tag{22}$$

Thus, the hypergraph pollution is computed by

$$\mathcal{HP}(\mathcal{E}) = \frac{1}{|\mathcal{E}|} \cdot |\mathcal{E}| \cdot 0 = 0 \tag{23}$$

$\square$

**Theorem 2.** *(Upper bound) The pollution score $HP(e_k)$ and $\mathcal{HP}(\mathcal{E})$ are upper-bounded by 1.*

*Proof.* Assuming that the nodes within a hyperedge are highly dissimilar from one another, the cosine similarity tends to 0. The pollution score of the hyperedge $e_k$ is formulated as

$$\begin{aligned} \text{HP}(e_k) &= 1 - \frac{1}{|e_k|^2} \sum_{i \in e_k} \left( \sum_{j \in e_k} \frac{\mathbf{X}_{i,:} \cdot \mathbf{X}_{j,:}}{||\mathbf{X}_{i,:}|| \cdot ||\mathbf{X}_{j,:}||} \right) \\ &= 1 - \frac{1}{|e_k|^2} \cdot |e_k| \cdot (|e_k| \cdot 0) \\ &= 1. \end{aligned} \tag{24}$$

$\square$

Thus, the hypergraph pollution is computed by

$$\mathcal{HP}(\mathcal{E}) = \frac{1}{|\mathcal{E}|} \cdot |\mathcal{E}| \cdot 1 = 1 \tag{25}$$

The above proof shows that both $\text{HP}(e_k)$ and $\mathcal{HP}(\mathcal{E})$ are bounded within the interval [0,1].

**Theorem 3.** *(Symmetry) For any two vertices $v_i$ and $v_j$, the $HP(e_k)$ is symmetric.*

*Proof.* Since the cosine similarity is invariant to the order of vertices, we have $\cos(\theta_{ij}) = \cos(\theta_{ji})$. That is to say, swapping the positions of $v_i$ and $v_j$ does not affect the value of $\text{HP}(e_k)$. Thus, the $\text{HP}(e_k)$ is symmetric. $\square$

**Theorem 4.** *(Monotonicity) The pollution score $HP(e_k)$ increases as intra-hyperedge heterogeneity increase.*

*Proof.* Let $s = \sum_{i,j \in e_k} \left( \frac{\mathbf{X}_{i,:} \cdot \mathbf{X}_{j,:}}{||\mathbf{X}_{i,:}|| \cdot ||\mathbf{X}_{j,:}||} \right)$, it is non-negative and $\text{HP}(e_k) = 1 - \frac{1}{|e_k|^2 s}$. Assuming the hyperedge size $|e_k|$ is fixed, the pollution score $\text{HP}(e_k)$ can be viewed as a function of a variable $s$, where $s$ denotes the aggregated similarity among node pairs within the hyperedge. Under this formulation, we have $\text{HP}(s) = 1 - \frac{1}{s}$. Taking the derivative to $s$, we obtain:

$$\frac{d}{ds}\text{HP}(s) = \frac{1}{s^2}. \tag{26}$$

This derivative is strictly positive when $s > 0$, indicating that $\text{HP}(s)$ is a monotonically increasing function as the intra-hyperedge similarity decreases. This supports the conclusion that $\text{HP}(e_k)$ grows with vertices heterogeneity within hyperedges. $\square$

A.3 EXPERIMENTAL ANALYSIS OF HP SCORE

Table 3: HP Score of Different Dynamic Hypergraph Methods on Nine Datasets.

| Method | Cora | Citeseer | PubMed | ModelNet40 | NTU2012 | Disease | Airport | Computers | Products | Avg. |
|--------|------|----------|--------|------------|---------|---------|---------|-----------|----------|------|
| DHGNN | 0.51 | 0.45 | 0.53 | 0.35 | 0.37 | 0.49 | 0.41 | 0.63 | 0.42 | 0.463 |
| TDHNN | 0.29 | 0.32 | 0.27 | 0.24 | 0.27 | 0.31 | 0.26 | 0.27 | 0.36 | 0.288 |
| DHHNN | 0.26 | 0.29 | 0.21 | 0.26 | 0.26 | 0.28 | 0.22 | 0.28 | 0.27 | 0.259 |
| HPHNN | 0.19 | 0.21 | 0.18 | 0.15 | 0.17 | 0.16 | 0.16 | 0.21 | 0.23 | 0.184 |

To further understand how the hyperedge pollution problem affects the performance of hypergraph neural networks (HGNNs), we analyze the quantitative behaviors of the proposed HP score and examine its impact on node-level representation learning. HGNNs suffer from degraded performance when high-pollution hyperedges dominate, since heterogeneous nodes are erroneously grouped. By leveraging the HP score to refine hyperedge structures adaptively, our HPHNN framework effectively mitigates the negative influence of hyperedge pollution. This results in more homogeneous hyperedges and improved robustness across datasets with varying degrees of HP score.

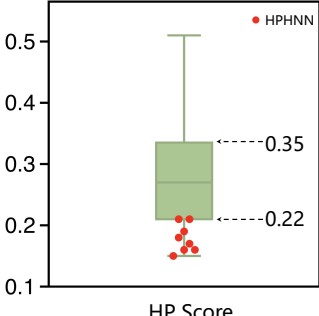

Figure 5: Boxplot of HP scores across methods, where Q1=0.22 and Q3=0.35; HPHNN remains mostly below Q1, indicating consistently low pollution.

**Three Levels of Hyperedge Pollution (HP) score**

Table 3summarizes the average HP score of the hypergraphs constructed by different dynamic hypergraph methods on nine benchmark datasets, providing a quantitative measure of intra-hyperedge

heterogeneity. According to the distribution of HP score shown in the boxplot (Fig. 5), we divide the values into three levels based on the lower and upper quartile points:

- **Low Hyperedge Pollution (HP score $< 0.22$):** The Score below the first quartile indicates that hyperedges are relatively homogeneous. Our method HPHNN consistently falls into this range, with values such as PubMed (0.18), ModelNet40 (0.15), NTU2012 (0.17), and Disease (0.16). This demonstrates that HPHNN effectively suppresses hyperedge pollution, maintaining clean and semantically coherent structures.

- **Moderate Hyperedge Pollution ($0.22 \leq$ HP score $< 0.35$):** The Score between the first and third quartiles reflects partial heterogeneity. Methods such as TDHNN and DHHNN typically lie in this range, e.g., Cora (0.29), Citeseer (0.32/0.29), and ModelNet40 (0.24/0.26). While these models mitigate pollution to some extent, the residual heterogeneity still weakens node-level discriminability, requiring further refinement.

- **High Hyperedge Pollution (HP score $\geq 0.35$):** The Score above the third quartile reveals severe hyperedge pollution. DHGNN, for example, produces high scores across most datasets, such as Cora (0.51), Citeseer (0.45), PubMed (0.53), and Computers (0.63). These high values indicate that many semantically unrelated nodes are grouped together, introducing substantial noise into representation learning. This highlights the necessity of HP-aware strategies.

Overall, as shown in Fig. 5, the HP scores of HPHNN are almost entirely below the first quartile (0.22), confirming its robustness in consistently reducing hyperedge pollution across benchmarks.

**Empirical Evidence**

**The Solution of the Hyperedge Pollution (HP) Challenge.** Our empirical study (Table 3) shows that the effectiveness of different hypergraph methods is strongly tied to their resulting HP scores. For instance, DHGNN produces consistently high HP scores, such as 0.51 on Cora, 0.45 on Citeseer, and 0.53 on Pubmed, indicating severe hyperedge pollution. In contrast, TDHNN and DHHNN generally yields moderate HP scores in the range of 0.22–0.35 (e.g., Citeseer: 0.32/0.29, Products: 0.36/0.27), reflecting partial heterogeneity. Finally, HPHNN achieves the lowest HP scores across all datasets, with values such as 0.18 on Pubmed, 0.15 on ModelNet40, and 0.16 on Disease, demonstrating its ability to construct clean and semantically coherent hyperedges. These results confirm that while traditional dynamic methods alleviate pollution to some degree, only our HP-aware refinement consistently reduces HP scores below the low threshold ($< 0.22$), thereby ensuring robust node-level learning across benchmarks.

## A.4    CONCLUSION

The pollution score provides a theoretically grounded and practically useful metric for evaluating the quality of hyperedges. It reflects both local consistency within each hyperedge and global structure across the hypergraph. Its boundedness, symmetry, and monotonicity make it suitable to be a regularization term in dynamic hypergraph refinement.

## B    EFFICIENCY ANALYSIS

Table 4: Training Time (ms/epoch) on Three Largest Datasets.

| Method | Pubmed | Computers | Products |
|---|---|---|---|
| GraphMVM | 9.6 | 8.9 | OOM |
| NormProp | 10.5 | 26.7 | 1136.5 |
| Polynormer | 145.1 | 47.9 | 3130.0 |
| FairGP | 36.1 | 25.4 | 1762.5 |
| TDHNN | 413.5 | 306.4 | 92465.7 |
| DHHNN | 383.0 | 237.7 | 78256.7 |
| **HPHNN** | 1837.5 | 1658.4 | 186729.4 |

To evaluate the computational efficiency and scalability of our method, we compare its training time and memory usage against several baselines on large-scale datasets. Tables 4 and 5 report the training time (ms/epoch) and peak GPU memory usage (GB) on the three largest datasets: PubMed, Computers, and Products. Compared to efficient baselines such as GraphMVM, NormProp, Polynormer, FairGP, TDHNN, and DHHNN, the proposed HPHNN method requires substantially more training time, especially on large datasets like Products. However, our peak GPU memory remains competitive and is often lower than other hypergraph-based methods (e.g., DHHNN).

The higher runtime mainly stems from three factors: (1) pollution score computation, which involves costly pairwise cosine similarity calculations within large hyperedges at each iteration; (2) dual attention mechanisms, applying attention from hyperedges to vertices and back, requiring dense matrix multiplications and softmax operations; and (3) clustering-based hypergraph reconstruction, where each iteration runs $k$-means clustering to form and refine hyperedges, adding overhead for large node sets.

Despite the extra cost, these design choices enable more coherent hyperedges with reduced pollution, adaptive incidence structures connecting nodes only to relevant hyperedges, and improved node representations, especially in noisy settings. This dynamic refinement leads to *stronger node classification performance* despite longer runtime.

In summary, while our method is slower than baselines due to pollution score computation, attention, and clustering, it maintains competitive GPU memory usage and achieves superior accuracy through refined hypergraph structures. Therefore, the additional computational cost is justified by the performance gains in node-level tasks.

Table 5: Peak GPU Memory Usage (GB) on Three Largest Datasets.

| Method | Pubmed | Computers | Products |
|---|---|---|---|
| GraphMVM | 0.89 | 0.69 | OOM |
| NormProp | 1.23 | 4.18 | 15.69 |
| Polynormer | 8.78 | 7.39 | 12.93 |
| FairGP | 1.48 | 0.99 | 6.29 |
| TDHNN | 1.14 | 2.65 | 12.96 |
| DHHNN | 7.04 | 4.41 | 14.83 |
| **HPHNN** | 1.13 | 3.31 | 10.42 |

# C DATASET DETAILS

## C.1 DATASET DESCRIPTIONS

Table 6: Benchmark Graph Datasets Overview.

| Dataset | Nodes | Edges | Features | Classes |
|---|---|---|---|---|
| Cora | 2,708 | 5,429 | 1,433 | 7 |
| Citeseer | 3,327 | 4,732 | 3,703 | 6 |
| Pubmed | 19,717 | 44,338 | 500 | 3 |
| ModelNet40 | 12,311 | 12,311 | 100 | 40 |
| NTU2012 | 2,012 | 2,012 | 100 | 67 |
| Disease | 1,044 | 2,086 | 1,000 | 2 |
| Airport | 3,188 | 37,261 | 4 | 2 |
| Computers | 13,752 | 245,861 | 767 | 10 |
| Twitch-PT | 1,912 | 62,598 | 128 | 2 |
| Leukemia | 4,651 | 6,362 | 4,608 | 3 |
| Products | 2,449,029 | 61,859,140 | 100 | 47 |

This work evaluates eleven benchmark datasets spanning diverse domains such as citation networks, 3D object recognition, epidemic simulation, transportation, e-commerce, social networks, and biomedical applications (Table 6). All datasets used in this work are publicly available. For ModelNet40 and NTU2012, we follow the same feature processing protocol as DHGNN (Jiang et al., 2019), and for Twitch-PT and Leukemia, we adopt the same data preprocessing procedure as in (Li et al., 2023). For all other datasets, we use their original raw features.

**Citation Networks (Cora, Citeseer, Pubmed):** These datasets model scientific publications as nodes linked by citations, with high-dimensional sparse features representing words or TF–IDF scores. Pubmed is notably larger and more complex.

**3D Object Recognition (ModelNet40, NTU2012):** Multi-view 3D object datasets where graphs are constructed from CNN features of object projections. ModelNet40 covers 40 classes; NTU2012 has 67 classes. **Disease :** A synthetic epidemic transmission network with tree structure; nodes encode susceptibility features and infection status, testing hierarchical and propagation modeling. **Airport (Flight Network):** Global airport network with nodes as airports connected by direct flights. Features include geographic and traffic attributes; labels relate to country population classes.

**Computers:** Product co-purchase network with bag-of-words product review features, aimed at classifying product categories.

**Twitch-PT:** A social network dataset of Twitch users from Portugal, where nodes are users and edges represent interactions. Features capture user profiles; classification is binary.

**Leukemia:** Biomedical graph dataset with 4,651 nodes representing cells, featuring high-dimensional gene expression data (4,608 features). Nodes are classified into three cell types, challenging models with rich biological complexity.

**Products:** A large-scale Amazon co-purchase network with over 2 million nodes and 61 million edges. Node features are 100-dimensional, with 47 product categories.

## C.2 DATA SPLITS

For Cora, Citeseer, and Pubmed, we adopt 50% training, 25% validation, and 25% testing splits as common in graph learning. ModelNet40 and NTU2012 use fixed 80/20 train/test splits without separate validation. Disease and Airport datasets are split 70%/15%/15%. Computers uses 60%/20%/20%. For Twitch-PT and Leukemia, we apply 70% training, 15% validation, and 15% testing splits to ensure robust evaluation. Products follows the official OGB benchmark splits. These protocols align with prior works (Chien et al., 2022; Zhou et al., 2023; Chen et al., 2023; Mei et al., 2025; Hu et al., 2020; Li et al., 2023).

## D IMPLEMENTATION DETAILS

We implement HPHNN using PyTorch. A grid search is conducted to determine optimal hyperparameters. The hidden dimension of node embeddings is selected from $\{16, 32, 64, 128, 256, 512\}$. The number of hypergraph convolutional layers is chosen from $\{1, 2\}$. For hyperedge construction, we search for the number of nodes $k_n$ that are most similar to the center of the hyperedge in the range of $\{5, 10, 15, 20\}$, and for each node, the number of hyperedges it is connected to, $k_e$, is chosen from $\{1, 2, 3, 4, 5, 10, 15, 20\}$. All models are trained using the Adam optimizer, and early stopping is applied based on the accuracy of the validation. All experiments were conducted on a workstation equipped with an NVIDIA RTX 4090 GPU and an Intel Xeon Gold 6226R CPU (2.90 GHz). Each was repeated with 10 different random seeds, and results are reported as mean ± standard deviation of accuracy.

## E ADDITIONAL EXPERIMENTS OF NODE-LEVEL TASK

To provide a more comprehensive evaluation of HPHNN beyond node classification, we further assess its performance on two additional node-level tasks: node regression and node clustering. Specifically, we replace the original loss function with the mean absolute error (MAE) loss in the node regression task.

Table 7 presents the regression results measured by MAE and mean squared error (MSE) on three widely used datasets: Chameleon, Crocodile, and Squirrel. Both MAE and MSE are metrics where lower values indicate better performance. Our method achieves the best performance on all three datasets and both metrics, outperforming strong baselines including NormProp and Polynormer. Specifically, on the chameleon dataset, our model yields the lowest MAE (0.521) and MSE (0.524), improving upon the previous best (NormProp: 0.540 MAE, 0.530 MSE). On crocodile, it further reduces the MAE to 0.427 and MSE to 0.436, surpassing Polynormer. On the squirrel dataset, while DHHNN achieves a competitive MAE of 0.695, our model slightly outperforms it with 0.694, also earning the lowest MSE (0.712). These results demonstrate that our dynamic hypergraph refinement effectively captures continuous label variations and structural semantics, making it well-suited for node-level regression tasks.

Table 7: Comparison of Node Regression Performance across Three Datasets.

| Method | Chameleon | | Crocodile | | Squirrel | |
|---|---|---|---|---|---|---|
| Metrics | MAE | MSE | MAE | MSE | MAE | MSE |
| UniG-Encoder | 0.600 | 0.610 | 0.480 | 0.500 | 0.750 | 0.770 |
| NormProp | 0.540 | 0.530 | 0.440 | 0.460 | 0.700 | 0.720 |
| Polynormer | 0.550 | 0.540 | 0.430 | 0.435 | 0.715 | 0.730 |
| FairGP | 0.560 | 0.550 | 0.445 | 0.455 | 0.705 | 0.720 |
| TDHNN | 0.570 | 0.560 | 0.450 | 0.460 | 0.710 | 0.725 |
| DHHNN | 0.575 | 0.565 | 0.455 | 0.465 | 0.695 | 0.710 |
| **HPHNN** | **0.521** | **0.524** | **0.427** | **0.436** | **0.694** | **0.712** |

Table 8 presents node clustering performance on the Cora and Citeseer datasets, evaluated using three metrics: normalized mutual information (NMI), adjusted rand index (ARI), and silhouette score (SS). Our model achieves the best results across all metrics on both datasets, consistently outperforming both traditional clustering (K-means) and state-of-the-art baselines. On Cora, our model achieves an NMI of 0.680, ARI of 0.590, and SS of 0.511, surpassing the previous best method (FairGP: 0.659 NMI, 0.573 ARI) by a clear margin. On Citeseer, it further improves performance with an NMI of 0.543, ARI of 0.537, and SS of 0.316, exceeding the previous best scores reported by DHHNN.

These improvements validate the effectiveness of dynamically updating hypergraph structure in capturing both global community information and local semantic consistency, leading to more accurate and separable clustering results.

Table 8: Comparison of Node Clustering Performance on Cora and Citeseer.

| Method | Cora | | | Citeseer | | |
|---|---|---|---|---|---|---|
| Metric | NMI | ARI | SS | NMI | ARI | SS |
| K-means | 0.202 | 0.125 | 0.015 | 0.211 | 0.163 | 0.009 |
| UniG-Encoder | 0.611 | 0.529 | 0.432 | 0.488 | 0.456 | 0.252 |
| NormProp | 0.634 | 0.542 | 0.465 | 0.496 | 0.463 | 0.265 |
| Polynormer | 0.652 | 0.567 | 0.460 | 0.512 | 0.479 | 0.280 |
| FairGP | 0.659 | 0.573 | 0.444 | 0.498 | 0.468 | 0.279 |
| TDHNN | 0.644 | 0.554 | 0.477 | 0.507 | 0.471 | 0.282 |
| DHHNN | 0.639 | 0.548 | 0.473 | 0.523 | 0.501 | 0.308 |
| **HPHNN** | **0.680** | **0.590** | **0.511** | **0.543** | **0.537** | **0.316** |

## F    FURTHER EXPERIMENTAL ANALYSIS FOR HPHNN

To better understand the computational behavior of HPHNN, we conduct a series of targeted experiments that isolate and quantify the major sources of computational overhead. Because HPHNN

repeatedly refines the hypergraph structure during training, our analysis centers on two key questions: (1) which components dominate the overall runtime, and (2) how the model's predictive performance changes when its parameter size is reduced. These examinations provide a clearer picture of the performance–cost trade-off of HPHNN.

**Component-wise runtime profiling.** We begin by examining the runtime of individual modules within HPHNN. The training pipeline is decomposed into clustering, HP-score computation, attention coefficient calculation, loss computation, and initial GNN embedding learning. As shown in Table 9, K-means clustering is by far the most expensive component, accounting for 68.0% of the total runtime. This arises because standard HPHNN executes K-means at every hypergraph refinement step, whereas HP-score computation and attention aggregation contribute only a small fraction of the overall cost.

Table 9: Runtime Breakdown of HPHNN Components.

| Operation | Time (ms) | Percentage |
|---|---|---|
| Total Time | 350.77 | 100% |
| HP Score | 56.92 | 16.2% |
| Loss Computation | 14.78 | 4.2% |
| K-Means Clustering | 238.61 | 68.0% |
| Node-to-edge Attention | 3.22 | 0.9% |
| Edge-to-node Attention | 31.10 | 8.9% |
| GCN Initial Embedding | 6.14 | 1.8% |

**Parameter-scaling evaluation.** We additionally examine how reducing model size affects predictive performance. By decreasing hidden dimensions, convolutional depth, dropout rate, and feature-transformation complexity, we obtain a compact 6.25M-parameter model—smaller than all dynamic hypergraph baselines. Despite this reduction, *HPHNN (reduced)* maintains strong accuracy across Cora, Citeseer, and Pubmed, as summarized in Table 10. These results indicate that HPHNN achieves an effective balance between parameter efficiency and prediction quality.

Table 10: Parameter Size And Node Classification Accuracy Comparison on Three Citation Datasets.

| Model | Parameters | Cora | Citeseer | Pubmed |
|---|---|---|---|---|
| DHGNN | 16,175,948 | 79.52% | 73.59% | 84.24% |
| TDHNN | $6,466,055$ | 88.67% | 79.57% | 87.48% |
| DHHNN | 7,203,856 | 89.29% | 80.92% | 89.47% |
| HPHNN (original) | 7,991,303 | 92.85% | 84.55% | 91.21% |
| HPHNN (reduced) | $6,248,647$ | 90.31% | 81.02% | 89.68% |

**Improving scalability by redesigning the clustering module.** Motivated by these findings, we revise HPHNN such that K-means is executed only once during initialization, and the resulting hypergraph is reused throughout training. A lightweight hyperedge split–merge strategy based on node similarity and hyperedge-center similarity further replaces repeated clustering. As reported in Table 11, the updated version, *HPHNN_new*, achieves nearly identical accuracy to the original model while reducing runtime by 6–8× across PubMed, Computers, and Products. This confirms that the redesign substantially enhances scalability.

**Ablation study on $\mathcal{S_H}$.** We further evaluate the contribution of each term through an ablation study on Cora, Citeseer, and Pubmed. As reported in Table 12, removing any component of $\mathcal{S_H}$ leads to consistent and notable performance degradation: Satur decreases accuracy by 2.7–2.8%, Silh by 2.2–2.3%, and HP by 3.0–3.5%. These results demonstrate that each term captures a different aspect of hypergraph quality, and all three are necessary for achieving optimal predictive performance.

**Relationship Between Hyperedge Pollution and Node-Level Performance** Understanding the relationship between hyperedge pollution (HP) and node-level predictive performance is essential for analyzing the behavior of hypergraph neural networks. The HP score quantifies semantic in-

Table 11: Runtime And Node Classification Accuracy Comparison on PubMed, Computers, And Products. HPHNN_new denotes the proposed improved version of HPHNN.

| Method | PubMed (Time/ACC) | Computers (Time/ACC) | Products (Time/ACC) |
|--------|-------------------|----------------------|---------------------|
| HGNN | 58.4ms / 79.39% | 52.3ms / 72.45% | 12418.6ms / 78.70% |
| DHGNN | 517.6ms / 79.52% | 461.3ms / 73.59% | 115684.6ms / 80.46% |
| TDHNN | 413.5ms / 87.48% | 306.4ms / 92.60% | 92465.7ms / 82.97% |
| DHHNN | 383.0ms / 89.47% | 237.7ms / 92.01% | 78256.7ms / 84.29% |
| HPHNN | 1837.5ms / 91.21% | 1658.4ms / 96.82% | 186729.4ms / 85.69% |
| HPHNN_new | 223.9ms / 91.33% | 222.3ms / 95.24% | 28382.3ms / 85.42% |

Table 12: Ablation Study on Three Components of The Hypergraph Quality Score.

| Model Configuration | Cora | Citeseer | Pubmed |
|---------------------|------|----------|--------|
| Full Model | 92.85% | 84.55% | 91.21% |
| w/o Satur | 90.12% | 81.73% | 88.47% |
| w/o Silh | 90.56% | 82.15% | 88.89% |
| w/o HP | 89.34% | 80.92% | 87.68% |

consistency within hyperedges; higher values indicate stronger heterogeneity, meaning that nodes grouped together share fewer meaningful relationships. Such inconsistency weakens the quality of the information aggregated during message passing and is therefore expected to negatively affect representation learning.

To study this relationship, we conduct a controlled intervention on the refined hypergraph. Noise is injected by inserting nodes sampled from other hyperedges, which systematically increases hyperedge heterogeneity while leaving the rest of the model unchanged. After each intervention, we recompute the HP score and evaluate node-level accuracy.

Table 13 summarizes the results. Increasing the amount of injected noise consistently raises the HP score and produces a monotonic decline in node-level performance across Cora, Citeseer, and Pubmed. Since hyperedge heterogeneity is the only factor modified between conditions, the observed trend demonstrates a direct relationship: higher HP scores correspond to poorer node-level predictive performance.

Table 13: Impact of Injected Noise on HP Scores And Node Classification Accuracy.

| Dataset | Noise | HP Score | HGNN | HyperGCN | HyperConv | HPHNN |
|---------|-------|----------|------|----------|-----------|-------|
| Cora | 0% | 0.19 | 82.3% | 81.6% | 88.7% | 92.85% |
| | 10% | 0.38 | 80.2% | 79.8% | 87.5% | 91.21% |
| | 20% | 0.45 | 77.6% | 77.1% | 84.2% | 89.57% |
| | 30% | 0.51 | 74.3% | 73.9% | 81.1% | 87.84% |
| Citeseer | 0% | 0.21 | 75.1% | 74.6% | 79.4% | 84.55% |
| | 10% | 0.36 | 74.5% | 73.8% | 77.9% | 83.26% |
| | 20% | 0.42 | 71.2% | 70.6% | 75.1% | 81.74% |
| | 30% | 0.49 | 67.8% | 67.1% | 72.8% | 79.97% |
| Pubmed | 0% | 0.18 | 89.2% | 85.4% | 88.7% | 91.21% |
| | 10% | 0.37 | 86.1% | 85.4% | 86.7% | 89.85% |
| | 20% | 0.44 | 84.4% | 82.8% | 84.3% | 88.33% |
| | 30% | 0.50 | 81.2% | 80.5% | 81.1% | 87.75% |

**Comparison of Different Similarity Metrics for HP Score** Cosine similarity has been widely adopted in various graph-based tasks for measuring semantic similarity between node embeddings. Previous studies, including "Similarity-Navigated Conformal Prediction for Graph Neural Networks

(NeurIPS 2024)", "Graph Neural Networks Need Cluster-Normalize-Activate to Better Avoid Oversmoothing (NeurIPS 2024)", and "Node Similarities under Random Projections (ICLR 2025)", have shown that cosine distance outperforms Euclidean and Manhattan distances in constructing similarity graphs for node embeddings. Our experimental results further validate its effectiveness in hypergraph neural networks, especially for node-level representation learning and the computation of HP scores.

The experimental results show that Cosine Similarity achieves the best performance in node classification across three datasets (Cora, Citeseer, and Pubmed). This supports its use in calculating the HP score, as it effectively captures semantic inconsistency and improves node-level task performance.

Table 14: Comparison of Different Similarity Metrics for HP Score in Node Classification Tasks

| Metric | Cora | Citeseer | Pubmed |
|---|---|---|---|
| Euclidean Distance | 91.02% | 82.13% | 88.95% |
| Manhattan Distance | 90.12% | 80.99% | 87.75% |
| Dot Product | 91.23% | 83.45% | 89.12% |
| Cosine Similarity | 92.85% | 84.55% | 91.21% |

**Ablation Studies on The Loss Functions in Eq. 18** The experimental results across all three datasets demonstrate that L2 distance achieves the highest node classification accuracy, while other distance metrics lead to significant performance degradation, supporting our selection of the L2-based loss function.

Table 15: Comparison of Different Distance Metrics in The Loss Function for Node Classification

| Distance Metric | Cora | Citeseer | Pubmed |
|---|---|---|---|
| L1 Distance | 88.72% | 80.15% | 87.03% |
| Cosine Distance | 89.21% | 80.43% | 87.28% |
| Negative Dot Product | 87.95% | 79.84% | 86.75% |
| Squared L2 Distance | 91.02% | 82.13% | 88.95% |
| L2 Distance | 92.85% | 84.55% | 91.21% |

**The Optimal Values of The Hyperparameter $\theta$ for Different Datasets** The table presents the optimal values of the hyperparameter $\theta$ for different datasets based on extensive hyperparameter tuning. The results show that $\theta$ only requires searching within a narrow range of [0.9, 0.95], demonstrating its low sensitivity.

Table 16: Optimal Hyperparameter $\theta$ for Different Datasets

| Dataset | Cora | Citeseer | Pubmed | ModelNet40 | NTU2012 | Disease | Airport | Computers | Twitch-PT | Leukemia | Products |
|---|---|---|---|---|---|---|---|---|---|---|---|
| $\theta$ | 0.90 | 0.90 | 0.95 | 0.93 | 0.91 | 0.94 | 0.90 | 0.92 | 0.91 | 0.94 | 0.95 |

The table presents the optimal values of the hyperparameter $\theta$ for different datasets based on extensive hyperparameter tuning. The results show that $\theta$ only requires searching within a narrow range of [0.9, 0.95], demonstrating its low sensitivity. These values were fixed per dataset, ensuring that the model performs optimally while maintaining stability across multiple datasets.

## G    PSEUDOCODE DESCRIPTION OF THE PROPOSED HPHNN

Here is the pseudocode outlining the key steps of our proposed HPHNN. It provides a clear overview of the main components and the iterative process involved.

## H    USE OF LARGE LANGUAGE MODELS

During the preparation of this manuscript, we used GPT-based large language models solely for *text polishing*, including grammar correction, readability improvement, and refinement of academic

---

**Algorithm 1:** HPHNN: Cluster-Dynamic Hypergraph Neural Network

---

**Input:** Feature matrix $\mathbf{X} \in \mathbb{R}^{n \times d}$;
Label matrix $\mathbf{Y}$;
Initial number of hyperedges $m^{(0)}$;
Attention parameters $k_n$, $k_e$;
Threshold $\theta$;
Maximum iterations $T$
**Output:** Predicted label matrix $\widehat{\mathbf{Y}}$

1 **Initialization:** Set $t \leftarrow 0$;
2 Obtain initial vertex embedding: $\mathbf{Z}^{(0)} \leftarrow \text{GNN}(\mathbf{X})$;
3 **while** $t < T$ **do**
4     **1. Hyperedge Construction via Clustering :**
5        Apply $k$-means to obtain $m^{(t)}$ hyperedges: $\mathcal{E}^{(t)} \leftarrow \text{KMeans}(\mathbf{Z}^{(t)}, m^{(t)})$;
6        **foreach** $e_k \in \mathcal{E}^{(t)}$ **do**
7           Compute hyperedge embedding: $\mathbf{u}_k \leftarrow \frac{1}{|e_k|} \sum_{i \in e_k} \mathbf{Z}_{i,:}^{(t)}$;
8        Stack: $\mathbf{U} \leftarrow [\mathbf{u}_1, \ldots, \mathbf{u}_{m^{(t)}}]$;
9     **2. Hyperedge Embedding Update via Attention :**
10        Compute attention: $\alpha_{e_i, v_j} \leftarrow \text{Softmax}(\mathbf{Q}_i \cdot \mathbf{K}_j^\top)$, where
11        $\mathbf{Q} = \mathbf{U}\mathbf{W}^Q, \quad \mathbf{K} = \mathbf{Z}^{(t)}\mathbf{W}^K$;
12        **foreach** *hyperedge* $e_i$ **do**
13           Select top-$k_n$ vertices: $\mathcal{N}_i \leftarrow \text{Top-}k_n(\{\alpha_{e_i, v_j}\})$;
14           Normalize attention: $\tilde{\alpha}_{e_i, v_j} \leftarrow \alpha_{e_i, v_j} / \sum_{j' \in \mathcal{N}_i} \alpha_{e_i, v_{j'}}$;
15           Update embedding: $\mathbf{u}_i \leftarrow \sum_{j \in \mathcal{N}_i} \tilde{\alpha}_{e_i, v_j} \cdot \mathbf{X}_{j,:} \mathbf{W}^V$;
16     **3. Vertex Representation Update :**
17        Reverse attention: $\alpha_{v_i, e_j} \leftarrow \text{Softmax}(\widehat{\mathbf{Q}}_i \cdot \widehat{\mathbf{K}}_j^\top)$, where
18        $\widehat{\mathbf{Q}} = \mathbf{Z}^{(t)}\widehat{\mathbf{W}}^Q, \quad \widehat{\mathbf{K}} = \mathbf{U}\widehat{\mathbf{W}}^K$;
19        **foreach** *vertex* $v_i$ **do**
20           Select top-$k_e$ hyperedges: $\hat{\mathcal{N}}_i \leftarrow \text{Top-}k_e(\{\alpha_{v_i, e_j}\})$;
21           Update incidence matrix $\mathbf{H}_{i,j} \leftarrow \alpha_{v_i, e_j}$ if $j \in \hat{\mathcal{N}}_i$, else 0;
22     Update vertex embedding: $\widehat{\mathbf{Z}} \leftarrow \mathbf{Z}^{(t)}\mathbf{W}_1 + \mathbf{H}\mathbf{U}\mathbf{W}_2$;
23     **4. Hypergraph Quality Evaluation and Adjustment :**
24        Compute score: $\mathcal{S}_{\mathcal{H}}^{(t)} = \text{Satur}(\mathcal{E}) + \text{Silh}(\widehat{\mathbf{Z}}, \mathbf{Y}) - \mathcal{HP}(\mathcal{E})$;
25        **if** $\mathcal{S}_{\mathcal{H}}^{(t)} > \theta$ **then**
26           $m^{(t+1)} \leftarrow m^{(t)} + 1$;
27        **else if** $\mathcal{S}_{\mathcal{H}}^{(t)} < \theta$ **then**
28           $m^{(t+1)} \leftarrow m^{(t)} - 1$;
29     **5. Prepare for Next Iteration:**
30        $\mathbf{Z}^{(t+1)} \leftarrow \widehat{\mathbf{Z}}$;
31     $t \leftarrow t + 1$;
32 **6. Final Prediction:**
33     Compute predicted label:
34     $\widehat{\mathbf{y}}_i = \log\left(\frac{\exp(\text{MLP}(\widehat{\mathbf{Z}}_i))}{\sum_{j=1}^n \exp(\text{MLP}(\widehat{\mathbf{Z}}_j))}\right)$;
35     Compute loss: $\mathcal{L} = \mathcal{L}_{\text{NLL}} + \mathcal{L}_{\text{VSC}} + \mathcal{L}_{\text{ESC}}$;
36 Backpropagate and update parameters;

---

writing style. All research ideas, model designs, experiments, and analyses were entirely conceived, implemented, and validated by the authors. The use of GPT did not affect the originality, technical contributions, or scientific conclusions of this work.

## I  ETHICS STATEMENT

This work adheres to the ICLR Code of Ethics. Our research does not involve human subjects, personal or sensitive data, or interventions with living beings. All datasets used in this study are publicly available and widely adopted in the research community. We carefully follow the corresponding licenses and cite the original sources. Our methods do not intentionally introduce bias or discriminatory outcomes, and we report results across multiple benchmarks to ensure robustness and fairness. The insights and algorithms developed are intended solely for academic research and pose no foreseeable risks of misuse or harm. All authors have reviewed and approved this submission in accordance with research integrity standards.

## J  REPRODUCIBILITY STATEMENT

We have made significant efforts to ensure the reproducibility of our work. All datasets used in this paper are publicly available, and details of the data preprocessing pipeline are provided in Appendix C. The full description of our model architecture and hyperparameter settings can be found in Appendix  G and Appendix D. We also include complete proofs of theoretical results in Appendix A. To further facilitate reproducibility, we provide our source code and configuration files in the supplementary materials. These resources collectively ensure that our experimental results and analyses can be replicated by the community.

