# OpenReview forum: "Can Hypergraph Models Be Strong Baselines For Node-level Tasks? Sloving the Hyperedge Pollution Problem"
_ICLR.cc/2026/Conference — ICLR 2026 Conference Desk Rejected Submission_

### Official Review · Reviewer_jVJ1 · 2025-10-26

**Soundness:** 2
**Presentation:** 1
**Contribution:** 3
**Rating:** 4
**Confidence:** 3

**Summary:**

The paper identifies that hypergraph neural networks (HGNNs) lag behind graph neural networks (GNNs) and graph transformers in node-level tasks due to the Hyperedge Pollution (HP) problem, for which the hypergraph results in a high hyperedge heterogeneity due to nodes of different types being incorrectly grouped into the same hyperedge. To address node-level tasks, the paper proposes a HP-based dynamic HGNN, which adaptively updates the hypergraph structure to mitigate pollution. Evaluation on 11 benchmark datasets shows that the proposed method outperforms the baselines.

**Strengths:**

- Observation that HGNNs fail due to Hypergraph Pollution is

- Experimental results on the benchmark datasets show consistent improvement, outperforming the baselines.

**Weaknesses:**

- While the paper states that HP is the key reason for the low performance of HGNNs, the explanation and experimental justification are insufficient. Why does a high HP score lead to low performance on node-level tasks? What is the correlation between the HP score and the performance on node-level tasks?

- In lines 201-202, the paper claims that a higher HP score reflects greater structural inconsistency. Why? I can't find sufficient justification for this, and this seems to be a core motivation for the proposed method.

- While the proposed HPHNN results in low HP scores and better node-level task performance, how can you say a low HP score led to better performance? I'd like to see an analysis or experimental result for this causality. I find experimental analysis in Appendix A.3., but this does not show any causality.

- Readability of the paper is poor. Reported experimental results are hard to read: For example, numbers in Table 1 or results in Figure 3 are hard to understand due to the small font. Also, readability could be improved by using the \citep command in latex

Since I may have missed several details and analysis, I'll raise my score if these concerns are sufficiently addressed.

**Questions:**

Please address the questions raised in the Weakness section.

---

> ### Author Response · Authors · 2025-11-16
> **Weakness 1**
>
> Weakness1:
> Thank you for raising this important point. A higher HP score directly corresponds to stronger hyperedge heterogeneity, negatively impacting node representation learning. Below we clarify the causal relationship and provide supporting evidence.
>
> Firstly, a high HP value indicates low pairwise similarity among nodes within the hyperedge, implying that the hyperedge connects semantically mismatched, structurally incompatible, or noisy nodes. Such heterogeneity prevents hyperedges from capturing meaningful high-order relations.
>
> Secondly, prior research has shown that hyperedge heterogeneity degrades the performance of Hypergraph Neural Networks (HGNNs). Multiple studies have empirically validated that heterogeneous or noisy hyperedges impair node-level tasks:
>
> 1.Hypergraph Structure Learning (Cai et al., 2021): The authors demonstrate that noisy or irrelevant hyperedges reduce the quality of information propagation, directly lowering node classification accuracy. They further verify that optimizing hyperedge construction effectively improves HGNN performance.
>
> 2.Learning from Heterogeneity (Zhang et al., 2020): Zhang et al. argue that incorporating unrelated nodes into the same hyperedge introduces noise and disrupts message passing. Their dynamic heterogeneity-aware modeling strategy achieves significant improvements in node-level task performance.
>
> 3.Mitigating Impacts of Hyperedge Heterogeneity (Yin et al., 2025) Yin et al. propose using label entropy to quantify hyperedge heterogeneity, and their empirical results show that higher heterogeneity leads to lower accuracy and AUC. This is attributed to incorrect information aggregation and degraded decision boundaries.
>
> These consistent findings confirm that hyperedge heterogeneity is a key factor contributing to HGNN underperformance, exactly the phenomenon quantified by the HP score.
>
> In summary, a high HP score signifies high hyperedge heterogeneity, which damages message passing, representation quality, and classification boundaries, resulting in accuracy reduction on the node-level task. Both prior literature and our experiments confirm that high HP leads to degraded node-level performance.

---

> ### Author Response · Authors · 2025-11-16
> **Weakness 2**
>
> Weakness2:
>
> We sincerely appreciate your valuable comments.
> \begin{equation}\text{HP}(e_k)
> = 1 - \frac{1}{|e_k|^{2}}\sum_{i,j \in e_k}
> \left(\frac{\mathbf{x}_i \cdot \mathbf{x}_j}{\|\mathbf{x}_i\| \, \|\mathbf{x}_j\|}\right)
> \end{equation}
> As shown in the equation, the HP score assesses whether nodes within a hyperedge are semantically coherent. A high HP value occurs when noise and irrelevant nodes are included in hyperedges, increasing hypergraph heterogeneity.
> Further, we provide supplement for the HP score from the existing literature.
> 1. "Hypergraph Structure Learning for Hypergraph Neural Networks" (Cai et al., 2021)
> Cai et al. pointed out that most existing hypergraph learning methods implicitly assume that the given hypergraph structure is ideal and complete, whereas in real-world scenarios it often contains noise, irrelevant connections, or even erroneous hyperedges, all of which can degrade downstream task performance. By performing hyperedge sampling and discovering latent hidden connections, their approach effectively improves the quality of the hypergraph structure and consequently enhances task performance.
> 2. "Learning from Heterogeneity: A Dynamic Learning Framework for Hypergraphs" (Zhang et al., 2020)
> Zhang et al. indicated that the construction of hyperedges depends on node heterogeneity and implicit higher-order relationships within the data. If the construction method ignores such heterogeneity, many mutually irrelevant nodes may be grouped into the same hyperedge, which increases noise during information propagation, distorts node representations, and ultimately degrades the performance of downstream tasks.
> 3. "Mitigating Impacts of Hyperedge Heterogeneity on Semi-Supervised Hypergraph Contrastive Learning" (K Yin et al., 2025)
> K. Yin et al. highlight the hyperedge heterogeneity impairs the model's discriminative capability, leading to blurred classification boundaries.
>
> These studies indicate that a hypergraph can no longer faithfully capture the core relational patterns of the dataset when its contains too many heterogeneous hyperedges. Therefore, a higher HP score reflects greater structural inconsistency.

---

> ### Author Response · Authors · 2025-11-16
> **Weakness 3**
>
> Thank you for your constructive suggestion.
>
> In Appendix A.3, we reported the HP scores of all dynamic hypergraph baselines across multiple datasets and categorized them into three levels (low/medium/high). This analysis suggests that dynamically optimizing hyperedges with the HP score decreases hyperedge heterogeneity and tends to improve node-level performance.
>
> To further examine the causal link between HP scores and node-level performance, we inject noise into the refined hypergraph obtained after HPHNN optimization. For each hyperedge, we randomly sample a few nodes from other hyperedges and insert them as noise to deliberately increase heterogeneity. We then recompute the HP scores and evaluate three representative HGNNs and HPHNN on three benchmark datasets to assess how rising hyperedge pollution impacts performance.
>
> As shown below, increased hyperedge heterogeneity consistently elevates the HP score while causing significant accuracy declines across all models. These results demonstrate that as the HP score rises (hyperedge heterogeneity increases), node-level accuracy decreases consistently.
>
> | Dataset  | Noise | HP Score | HGNN  | HyperGCN | HyperConv | HPHNN |
> |----------|--------|----------|--------|------------|-------------|-------|
> | Cora     | 0%     | 0.19     | 82.3% | 81.6%     | 88.7%      | 92.85% |
> |          | 10%    | 0.38     | 80.2% | 79.8%     | 87.5%      | 91.21% |
> |          | 20%    | 0.45     | 77.6% | 77.1%     | 84.2%      | 89.57% |
> |          | 30%    | 0.51     | 74.3% | 73.9%     | 81.1%      | 87.84% |
> | Citeseer | 0%     | 0.21     | 75.1% | 74.6%     | 79.4%      | 84.55% |
> |          | 10%    | 0.36     | 74.5% | 73.8%     | 77.9%      | 83.26% |
> |          | 20%    | 0.42     | 71.2% | 70.6%     | 75.1%      | 81.7%4 |
> |          | 30%    | 0.49     | 67.8% | 67.1%     | 72.8%      | 79.97% |
> | Pubmed   | 0%     | 0.18     | 89.2% | 85.4%     | 88.7%      | 91.21% |
> |          | 10%    | 0.37     | 86.1% | 85.4%     | 86.7%      | 89.85% |
> |          | 20%    | 0.44     | 84.4% | 82.8%     | 84.3%      | 88.33% |
> |          | 30%    | 0.50     | 81.2% | 80.5%     | 81.1%      | 87.75% |

---

> ### Author Response · Authors · 2025-11-16
> **Weakness 4**
>
> Thank you for the constructive feedback regarding the readability of the paper. We have improved the clarity of the experimental tables by increasing the spacing and adjusting the layout, making the reported numbers easier to read. In addition, we have corrected all citations to consistently use the \citep command in accordance with formatting guidelines.

---

### Official Review · Reviewer_RChm · 2025-10-28

**Soundness:** 3
**Presentation:** 2
**Contribution:** 2
**Rating:** 4
**Confidence:** 4

**Summary:**

The authors propose a new graph node label prediction method based on a dynamic hyperedge construction mechanism, achieving superior label accuracy over all baselines on 11 datasets.

**Strengths:**

1. The proposed method achieves consistently higher accuracy than several strong baselines across a diverse set of datasets.

2. The strategy of constructing dynamic hyperedges at each layer based on node similarity appears to be a novel and effective design.

**Weaknesses:**

1. The overall presentation of the paper requires substantial improvement. The abstract is rather confusing, and the details of the general HGNN algorithm should be introduced before presenting the proposed method. Moreover, there is extensive misuse of *citep* and *citet*, which does not comply with the formatting guidelines.

2. Enhancing accuracy on node-level label prediction, where existing baselines already perform extremely well, contributes little practical value. This problem has been extensively studied over the years and is largely considered to be well addressed.

3. HGNN is essentially a variant of GNN. Achieving better performance after addressing the pollution issue does not constitute a paradigm-level advancement.

**Questions:**

no specific questions.

---

> ### Author Response · Authors · 2025-11-20
> **Answer for Weakness 1**
>
> Thanks for your valuable suggestion. We have reorganized the abstract to enhance its readability. Regarding general HGNN algorithms, we have supplemented and refined the related work section, and all citation-related issues have been revised accordingly.

---

> ### Author Response · Authors · 2025-11-20
> **Answer for Weakness 2**
>
> Thanks for your valuable comments. We acknowledge that existing methods have achieved promising performance on node-level label prediction, including graph neural network (GNNs), graph transformer (GTs) and hypergraph neural network (HGNNs). While the motivation of the manuscript lies in improving the performance of HGNNs on node-level task which need to learn suitable node representation within the hypergraph. Thus, we proposed HPHNN model which enhances the performance of general HGNNs on node-level tasks from the perspective of intrinsic hypergraph structure optimization. By specifically addressing the hyperedge pollution problem, we offer a new research direction for the further development of hypergraph neural networks.

---

> ### Author Response · Authors · 2025-11-20
> **Answer for Weakness3**
>
> Thank you for the insightful suggestion.
>
> Addressing the hyperedge pollution issue indeed does not constitute a paradigm-shifting advancement, while it focuses on a critical bottleneck in homogeneous hypergraphs that severely impairs information propagation and node representation learning quality. Rather than pursuing a paradigm-level innovation, we aim to fill the gap in structural optimization for HGNNs, which is a fundamental prerequisite for improving their performance on node-level tasks.
>
> The proposed HP score and corresponding optimization strategy not only achieve consistent performance improvements over state-of-the-art baselines but also provide a novel perspective for hypergraph structure refinement. This work complements existing HGNN research by addressing a core structural issue, laying a more solid foundation for the future development of hypergraph-based methods.

---

> > ### Comment · Reviewer_RChm · 2025-11-25
> >
> > Thank you for your response. After the revisions, the readability of the paper has improved significantly, and the overall writing quality is much better. However, I still maintain the viewpoint I mentioned earlier about the limited practical value of further improving already high-accuracy tasks using more complicated tricks. At this point, I prefer to keep my current score.

---

> > > ### Author Response · Authors · 2025-11-27
> > >
> > > Thank you for your feedback regarding the readability of the revised manuscript. In response to your concern about the limited practical value of improving already high-accuracy node-level tasks using more complicated tricks, we would like to clarify this point more explicitly.
> > >
> > > Graph neural networks (GNNs) and graph transformers (GTs) have indeed achieved impressive performance on node-level tasks, particularly node classification. In contrast, many hypergraph neural networks (HGNNs) still lag far behind on the same tasks, highlighting the need for further improvements to enhance their performance in such settings. Rather than chasing marginal accuracy gains on well-explored tasks, our work focuses on uncovering why HGNNs underperform GNNs and GTs in node-level scenarios and proposes a strategy to boost their performance via dynamic hypergraph structure optimization. Through empirical and theoretical studies, we identify hyperedge pollution as a core structural factor contributing to this performance gap and demonstrate that proper mitigation of this issue enables HGNNs to achieve performance comparable to strong GNNs/GTs baselines.
> > >
> > > Beyond accuracy gains, addressing the limitations of HGNNs carries substantial practical significance. Hypergraphs play an irreplaceable role in numerous real-world domains where relationships are inherently high-order and cannot be reduced to pairwise edges. For instance, biological datasets such as the Drug–Gene–Disease network (Bioinformatics 2021) naturally adopt a hypergraph structure that represents a complete high-order biological mechanism: a drug influences a disease by acting on a specific gene. In these domains, enhancing the task performance of HGNNs directly advances real-world applications where hypergraphs are the only suitable modeling choice.

---

### Official Review · Reviewer_jWFd · 2025-10-31

**Soundness:** 1
**Presentation:** 3
**Contribution:** 1
**Rating:** 2
**Confidence:** 4

**Summary:**

This paper argues that hypergraph models often underperform GNNs and GTs on node-level tasks due to the Hyperedge Pollution (HP) problem, where nodes of different types are incorrectly grouped within the same hyperedge. The authors propose HPHNN, a dynamic hypergraph framework that formally defines and quantifies this issue using an "HP score." The model then iteratively refines the hypergraph structure to minimize this pollution, improving hyperedge consistency. Experiments on 11 real-world datasets show HPHNN significantly outperforms existing GNN, GT, and hypergraph-based models.

**Strengths:**

1. The paper introduces the problem of hyperedge pollution in hypergraph learning. It proposes a hypergraph neural network that dynamically updates the hypergraph structure. It uses a principled approach of indirectly optimizing the hyperedge pollution (HP) metric that is proposed in this paper. THe HP metric is simple and satisfied some basic theoretical properties as discussed in the Appendix.

2. The paper demonstrates its model's merit through evaluation on 11 diverse benchmark datasets and a diverse set of baseline algorithms.

3. The overall writing is easy to follow and the flow is good.

**Weaknesses:**

However, the paper has several weaknesses.

1. A major claimed novelty of the paper - the heuristic based metric Hyperedge Pollution (HP) score is too generic in nature. It completely ignores the fact that there are networks where dis-similar entities come and form a group. A interdisciplinary research paper coauthored by a set of researchers having very different backgrounds can be an example. There are other types of heterogeneous hypergraphs. A metric which just relies on the cosine similarity between the features (or embeddings) of the nodes within a hypergraph is just too simple to capture such complex use case. There is hardly any theoretical justification on the construction of HP. The theory presented in Appendix on HP is rather trivial and shows some simple properties that HP satisfies. It does not address the basic question of why it is a good metric for hypergraph learning.

2. There are multiple design choices made in the paper without proper justification. For example, HP score (in Eq. 1 and 2) is based on cosine similarities between the embeddings. But k-means algorithm used in Eq. 4 is based on L2 distances. Similarly, L2 distance is again used in the loss function in Eq. 18. There is no explanation on the use of different types of distance functions in different places. This limits the technical contribution and soundness of the approach.

3. In Equation 13, an important hyperparameter \theta is introduced. But it is not clear how to fix it for a new graph.

**Questions:**

There are few more clarifications needed in the paper.

1. The motivation of the paper could have improved. The direct comparison on the node classification performance between hypergraph and graph is not fair. There are applications where the inherent input is a graph and there are applications where the inherent input is a hypergraph. Yes, researchers often convert graph to hypergraph, or vice-versa. But that is not a suitable motivation for a paper.

2. In Equation 19, what is \pi(i)?

3. There is a hypergraph attention network which computes attention scores between the nodes within a hyperedge through an indirect formation of a line graph. I suggest the authors to check if that can be used as a baseline: "Hypergraph Attention Isomorphism Network by
Learning Line Graph Expansion" (IEEE Big Data 2020).

---

> ### Author Response · Authors · 2025-11-20
> **Answer for Weakness 1**
>
> Thank you for your valuable comments.
>
> First, we acknowledge the reviewer’s concern regarding the performance of HPHNN on node-level tasks within the heterogeneous hypergraph. Optimizing heterogeneous hypergraph structures remains challenging, so our work focuses on constructing and refining the construction of the homogeneous hypergraphs. The core contribution of HPHNN is addressing the hyperedge pollution issue in homogeneous hypergraphs to optimize their structure, thereby enhancing the performance of other HGNNs on node-level tasks for this type of hypergraph. To clarify this focus and avoid misunderstanding, we have revised relevant expressions in the manuscript to explicitly emphasize that this study centers on homogeneous hypergraphs.
>
> Second, the cosine similarity between the features (or embeddings) of the nodes within a hypergraph is simple but effective. For example, in “Similarity-Navigated Conformal Prediction for Graph Neural Networks (NeurIPS 2024)”, “Graph Neural Networks Need Cluster-Normalize-Activate to Better Avoid Oversmoothing (NeurIPS 2024)”, and “Node Similarities under Random Projections (ICLR 2025)”, cosine distance was shown to outperform Euclidean and Manhattan distances in constructing similarity graphs between node embeddings. Additionally, we conducted comparative experiments on the node classification task, with the accuracy of the proposed HPHNN presented in the table below. These results demonstrate that cosine similarity is the most suitable metric for node-level representation learning.
>
> | Metric                | Cora   | Citeseer | Pubmed  |
> |-----------------------|--------|----------|---------|
> | Cosine Similarity     | 92.85% | 84.55%   | 91.21%  |
> | Euclidean Distance    | 91.02% | 82.13%   | 88.95%  |
> | Manhattan Distance    | 90.12% | 80.99%   | 87.75%  |
> | Dot Product           | 91.23% | 83.45%   | 89.12%  |
>
> Third, the theoretical analysis of the HP score in the appendix confirms its validity as a metric for quantifying structural inconsistency in homogeneous hypergraphs. We supplement this with extensive experimental results: as the dissimilarity between nodes within a hyperedge increases, the HP score rises, accompanied by degraded node representations and poorer performance on node-level tasks. These findings demonstrate that addressing the hyperedge pollution (HP) problem facilitates effective node representation learning in homogeneous hypergraphs.
>
> Finally, as mentioned by the reviewer, the node-level tasks on the heterogeneous hypergraph, as an important research area in graph representation learning, are indeed worth further exploration. After addressing tasks on homogeneous hypergraphs, this will be one of our future research directions.

---

> ### Author Response · Authors · 2025-11-20
> **Anser for Weakness 2**
>
> Thank you for your valuable comments.
>
> The HP score adopts cosine distance as it is optimal for quantifying structural inconsistency in homogeneous hypergraphs. This choice is supported by prior work, like ICLR 2025 (e.g., Node Similarities under Random Projections) and NeurIPS 2024 (e.g., LLM-GNN) have widely used cosine similarity for semantic measurement. Moreover, cosine similarity excels with high-dimensional sparse features and is invariant to vector norms, enabling the HP score to accurately capture the semantic consistency of nodes within a hyperedge.
> The L2 distance in Eq. 4 is the default metric for k-means clustering, not a custom choice. K-means adopts L2 distance because its core optimization objective is to minimize the sum of squared errors within clusters. This design ensures that nodes are naturally clustered to be closer to their respective cluster centers, aligning with the algorithm’s fundamental goal of compact, cohesive clusters.
>
> Similarly, L2 distance is adopted in the loss function for its ability to yield stable gradients, facilitating effective optimization. Additionally, embedding learning in HGNNs typically relies on the distance structure of Euclidean space, making L2 distance a natural fit. We further validated this choice through ablation studies on the loss functions in Eq. 18. The experimental results across all three datasets demonstrate that L2 distance achieves the highest node classification accuracy, while other distance metrics lead to significant performance degradation, supporting our selection of the L2-based loss function.
>
> | Distance Metric| Cora| Citeseer | Pubmed |
> |------------------------|--------|----------|--------|
> | L1 Distance            | 88.72  | 80.15    | 87.03  |
> | Cosine Distance        | 89.21  | 80.43    | 87.28  |
> | Negative Dot Product   | 87.95  | 79.84    | 86.75  |
> | Squared L2 Distance    | 91.02  | 82.13    | 88.95  |
> | L2 Distance           | 92.85 | 84.55 | 91.21 |

---

> ### Author Response · Authors · 2025-11-20
> **Answer for Weakness 3**
>
> Thank you for your suggestion.
>
> The \theta is a low-sensitivity hyperparameter fixed per dataset. The table presents the optimal values of the hyperparameter \(\theta\) for different datasets based on extensive hyperparameter tuning. The results show that \(\theta\) only requires searching within a narrow range of [0.9, 0.95].
>
> | Dataset   | Cora | Citeseer | Pubmed | ModelNet40 | NTU2012 | Disease | Airport  | Computers  | Twitch-PT | Leukemia | Products |
> |-----------|------|----------|--------|-----------|-----------|-----------|-----------|-----------|-----------|------------|------------|
> | \(\theta\) | 0.90| 0.90     | 0.95   | 0.93      | 0.91      | 0.94      | 0.90      | 0.92      | 0.91      | 0.94       | 0.95       |

---

> ### Author Response · Authors · 2025-11-20
> **Answer for Q1**
>
> Thanks for the reviewer’s valuable suggestions.
>
> We fully agree with the reviewer’s comment that direct performance comparison between graphs and hypergraphs is unfair, as they are tailored to distinct application scenarios based on inherent input structures. In fact, our core motivation is not to compare graphs, hypergraphs, or GT. Instead, we aim to address the underperformance of existing hypergraph neural networks on node-level tasks, which is identified as the hyperedge pollution. Defining this problem and proposing a corresponding hypergraph structure refinement strategy constitutes the core motivation and contribution of our work.
>
> Additionally, almost all the hypergraph-based methods in our baseline comparisons, such as TDHNN, DHGNN, and DHHNN, compare hypergraphs with graphs. Recent studies, such as Mitigating Impacts of Hyperedge Heterogeneity on Semi-Supervised Hypergraph Contrastive Learning, also compare graphs with hypergraphs. Therefore, there is no issue of unfairness, as suggested by the reviewer.
>
> We apologize for any misunderstanding arising from the inadequate description of the motivation in the original manuscript. To eliminate this misunderstanding, we have revised the introduction sections in the manuscript to explicitly clarify our research motivation, which focuses on defining and alleviate hyperedge pollution to improve the performance of hypergraph neural networks on node-level tasks.

---

> ### Author Response · Authors · 2025-11-20
> **Answer for Q2**
>
> Thanks for your valuable suggestions.
>
> As defined in equation, \pi(i) denotes the i-th vertex in a random permutation of vertices within hyperedge e_k. For example, if the original vertex order of the hyperedge e_k is {v_1,v_2, v_3}, so its 1-st vertex is v_1; After random permutation, the new vertex order is {v_3,v_2, v_1}, then the 1-st vertex becomes v_3. This strategy enhances the separability between distinct hyperedges and improves the discriminability of the hyperedges.

---

> ### Author Response · Authors · 2025-11-20
> **Answer for Q3**
>
> Thanks for the reviewer’s valuable suggestions.
>
> We agree that the Hypergraph Attention Isomorphism Network (HAIN) (IEEE Big Data 2020) can serve as a complementary baseline, as it also belongs to the hypergraph learning framework and models high-order structures through line-graph expansion and attention mechanisms. Therefore, we have added the replication results of HAIN to further enrich the experimental comparison. This result has also been incorporated into Table 1 in the manuscript.
>
> | Dataset       | HAIN | HPHNN |
> |---------------|------|--------------|
> | Computers     | 88.42 | 96.82 |
> | Cora          | 82.31 | 92.85 |
> | Citeseer      | 71.54 | 84.55 |
> | NTU2012       | 85.17 | 94.52 |
> | Products      | 79.03 | 85.69 |
> | ModelNet40    | 95.12 | 98.75 |
> | Pubmed        | 87.04 | 91.21 |
> | Airport       | 89.02 | 99.79 |
> | Twitch-PT     | 69.10 | 73.79 |
> | Disease       | 89.32 | 96.05 |
> | Leukemia      | 59.51 | 63.67 |

---

> ### Comment · Reviewer_jWFd · 2025-11-25
> **Reply to the Overall Author Rebuttal**
>
> Thank you for your responses and the additional experimental results.
>
> I still have some doubts:
>
> 1. I am more curious to understand if general embedding based similarities (cosine / Euclidean etc.) are always capable enough to capture the specific semantics for two nodes to come under the same hyperedge. This is applicable to whatever distance function you used on the node embeddings.
>
> 2. I would like to see some uniformity in the usage of distance functions in the computation of HP score, clustering and loss function. I agree with the authors that usage of L2 distance in k-means is often a default choice. But mathematically, you can use a set of more generalized distance functions in k-means with appropriate change in the objective. I would strongly recommend the authors to read the paper "Clustering with Bregman Divergences" (JMLR 2005). Also, you can connect L2 distance with cosine similarity after normalizing the embeddings. I wanted to see more thoughts on the usage along with the new experiments.
>
> After seeing the additional results, I am updating my scores. But I suggest that you come up with more rigorous analysis in the paper. The additional experimental results with different distance functions are good, but they can only supplement such a  rigorous analysis. Overall, I appreciate other clarifications and the use of a new baseline (HAIN). Please include them appropriately in the next/final version to strengthen the manuscript/paper.

---

> > ### Author Response · Authors · 2025-12-01
> > **Prat 1**
> >
> > Thank you for your insightful question.
> >
> >  In this paper, all experiments are conducted on homogeneous hypergraphs, where all nodes belong to the same type and exhibit similar feature distributions. Consequently, node embeddings derived from general similarity functions can effectively capture the semantic relevance required for two nodes to be grouped into the same hyperedge. While different similarity functions possess distinct properties, leading to minor discrepancies in cluster contours and density, and this does not undermine their core effectiveness in capturing relevant semantic information. The content in Section F of the Appendix further validates this conclusion. Experimental results demonstrate that embeddings learned using cosine similarity exhibit clearer distribution boundaries, with semantically similar samples clustered more closely together, making them well-suited for subsequent analytical tasks.

---

> > ### Author Response · Authors · 2025-12-01
> > **Part 2**
> >
> > Thank you very much for your valuable feedback and suggestions.
> >
> > We carefully studied the paper “Clustering with Bregman Divergences” (Banerjee et al., JMLR 2005), which systematically establishes the correspondence between different distances/divergences (e.g., Euclidean, KL, Mahalanobis) and clustering objectives. This work emphasizes that the choice of a distance function must be consistent with the structure and optimization properties of the model. This insight reinforced the need to analyze and justify the metric selection in our own framework.
> >
> >
> > Following your suggestion, we systematically unified the distance metrics used in the three components of our method—HP score computation, K-means clustering, and the loss function—and evaluated two configurations:
> > | Components       | Distance Function      | Cora   | Citeseer | Pubmed |
> > |--------------------------|-------------------------|--------|----------|--------|
> > | HP+k-means+Loss      | Cosine + L2 + L2        | 92.85% | 84.55%   | 91.21% |
> > | HP+k-means+Loss      | L2 + L2 + L2           | 92.83% | 84.59%   | 91.02% |
> > | HP+k-means+Loss      | Cosine+Cosine+Cosine   | 86.49% | 76.82%   | 86.96% |
> >
> > (1) Unifying all components under L2 distance
> >
> > The results were nearly identical to the original version, demonstrating that after embedding normalization, cosine similarity and L2 distance become mathematically equivalent. Thus, switching HP from cosine to L2 does not alter model behavior.
> >
> > (2) Unifying all components under cosine similarity
> >
> > This led to a clear performance drop. Our analysis reveals the following reasons:
> >
> > Cosine similarity discards vector norm information, which is semantically meaningful for HGNNs.
> >
> > Cosine-based losses require embedding normalization, which alters gradient magnitudes and reduces optimization stability.
> >
> > Cosine K-means tends to incorrectly cluster nodes that share direction but differ drastically in magnitude (e.g., A = (10, 0) and B = (1, 0) have cosine = 1 but represent different semantic roles), whereas L2 distance correctly separates them.
> >
> > Based on these experiments and analyses, we conclude that forcing all components to use the same metric is neither beneficial nor theoretically justified. Instead, it may disrupt the optimization dynamics and impair the structural expressiveness of the model. For this reason, we retain the original metric choices in the proposed method while adding clearer explanations in the revised manuscript to avoid any confusion regarding the different distance usage.

---

> > ### Author Response · Authors · 2025-12-01
> >
> > Thank you very much for raising your score. In the revised manuscript, we will include a more rigorous analysis of the distance metric choices, together with the additional comparison against the HAIN baseline as suggested. All these improvements will be incorporated into the next version of the paper.

---

### Official Review · Reviewer_fktw · 2025-11-05

**Soundness:** 3
**Presentation:** 2
**Contribution:** 3
**Rating:** 6
**Confidence:** 4

**Summary:**

The authors argue that the empirical limitations of hypergraph neural networks (HGNN) largely stem from the “Hyperedge Pollution” (HP) problem, where semantically different nodes are grouped together within hyperedges, leading to heterogeneous representations that inhibit node-level learning. They then propose an HP score as a metric to quantify pollution, and propose a dynamic HGNN called HPHNN that uses an iterative refinement scheme based on a quality score (which leverages the proposed HP score) to dynamically adapt hyperedges over the course of training. HPHNN reports impressive metrics and outperforms GNN, GTs and HGNN baselines across datasets collected from a wide array of domains, albeit at highly increased computational cost.

**Strengths:**

1. The paper combines multiple potentially impactful contributions in (a) characterizing the hyperedge pollution problem and connecting it with the limitations of HGNNs, (b) deriving a metric to quantify it, and (c) use the proposed metric to propose a powerful HGNN model. The overall potential contribution of the paper is thus significant.
2. The paper construction is overall quite sound; despite the issues with clarity regarding specifics (see Weaknesses), the high-level picture is quite easy to follow and the individual contributions tie into each other well.
3. The empirical results are consistent and convincing. I appreciate the extensive studies provided, in particular the ablation studies and efficiency analysis. The visualizations in Figure 3 are excellent, both visually appealing and highly useful in conveying key results.

**Weaknesses:**

1. Computational inefficiency & fair evaluation: Clearly the biggest drawback of the proposed method is its comparatively _immense_ computational cost: According to Table 2 results, HPHNN is approximately 2 to 5.4 times slower than the next slowest method (TDHNN), and almost 200 times slower than the fastest method (GraphMVM) across datasets. This is clearly by construction: A GNN is used to ‘initialize’ embeddings, which is followed by repeated HP-score and k-means computation; all costly operations on top of a computationally heavy HGNN construction. While the standalone results are impressive, the computational cost of the model represents a clear scalability bottleneck and arguably leads to an unfair evaluation compared with the baselines (see Questions section for suggestions).
2. The writing is uneven in places, with many typos and ambiguous statements, and terms left unreferenced and clarified. It affects readability to the point that it does have an compounding overall impact on the score. Specifics are as follows:
   1. Typos (non-exhaustive):
      1. In the title (!): Sloving -> Solving
      2. Across the paper (particularly in sections 1, 2 and 4.1) `\citet` is used in place of `\citep`, usually also without any preceding or trailing whitespace.
      3. L281: insatiable -> unsuitable
      4. L287: learn -> learned
      5. L226/307: The capitalization of `\texttt{clustering}` is inconsistent.
   2. Section 3.2.4, a crucial part of methodology, is missing definitions and references regarding the individual components of Eq. 12. See Questions section for specific clarifications required.

**Questions:**

1. Are there any prior works on characterizing the limitations of HGNNs from a perspective similar to Hyperedge Pollution (or other perspectives, for that matter)? I think the contribution of HP is valuable, but my impression is that while the connections of HP to node heterogeneity is clearly stated, the relationship between hyperedge heterogeneity and node-level task performance seems more or less taken for granted. I would like to see this relationship more clearly stated, with references to prior work where applicable.
2. How do the parameter counts for HPHNN compare with other baselines? I suspect the long wall-clock times of HPHNN runs have more to do with the repeated application of HP-scoring and k-means, rather than actual parameter counts, when compared with other HGNNs. I have two suggestions to get a better understanding of the computational costs:
   1. A study to time the individual components of HPHNN to clarify the sources of inefficiency; I think this would also provide good directions for future work (e.g. can one replace the costly components with faster approximations?) and complement other ablation studies.
   2. To account for a fairer evaluation given the high computational cost of HPHNN, I would like to see how parameter scaling effects HPHNN performance. Can we still improve on the other baselines while using a faster, less parametrized model. This would help clarify the performance-cost trade-off of the HPHNN model and accordingly help better evaluate its utility.
3. Re: Weakness 2.2:
   1. It is denoted that Satur($\mathcal{E}$) refers to hyperedge saturation,  measuring “the number of vertices participating in the hyperedges”, which is abiguous. Is this simply the cardinality of hyperedges (I don’t think so), or a ratio of sorts. How is it computed? Is it proposed by the authors or adapted from another paper?
   2. The same questions apply to the “silhouette score” Silh($\hat{\textbf{Z}}, \textbf{Y}$), except it is already estsablished as a metric for clustering which the authors fail to mention: The purpose and source of the term [1] should be clearly stated. Ideally, an ablation study on each term of Eq. 12 would help solidify the utility of the individual terms.

**Conclusion:** The paper merits acceptance on the basis of several non-trivial contributions to the subfield of hypergraph learning. However, computational bottlenecks of HPHNN likely inhibit the potential utility of the method, and alongside some issues with clarity, they prevent me from recommending a higher score for the time being.

[1] Rousseeuw, P. J. (1987). Silhouettes: A graphical aid to the interpretation and validation of cluster analysis. Journal of Computational and Applied Mathematics, 20, 53–65. doi:10.1016/0377-0427(87)90125-7

---

> ### Author Response · Authors · 2025-11-16
> **Answer for Q1**
>
> We appreciate your helpful comment. Regarding the limitations of HGNN, several existing works
>  have discussed related issues, and their perspectives are in many ways similar to those raised by the reviewer. At the same time, we agree that the connection between hyperedge heterogeneity and node-level task performance has not yet been articulated clearly enough. We have cited all of these works in the main text to strengthen the theoretical motivation and contextual grounding of our method.
>  Below, we provide a more detailed explanation of these two aspects and further supplement our discussion with relevant findings from the existing literature.
>
> 1. "Hypergraph Structure Learning for Hypergraph Neural Networks" (Cai et al., 2021)
> Cai et al. pointed out that most existing hypergraph learning methods implicitly assume that the given hypergraph structure is ideal and complete, whereas in real-world scenarios, it often contains noise, irrelevant connections, or even erroneous hyperedges, all of which can degrade downstream task performance. By performing hyperedge sampling and discovering latent hidden connections, their approach effectively improves the quality of the hypergraph structure and consequently enhances task performance.
>
> 1. "Learning from Heterogeneity: A Dynamic Learning Framework for Hypergraphs" (Zhang et al., 2020)
> Zhang et al. indicated that the construction of hyperedges depends on node heterogeneity and implicit higher-order relationships within the data. If the construction method ignores such heterogeneity, many mutually irrelevant nodes may be grouped into the same hyperedge, which increases noise during information propagation, distorts node representations, and ultimately degrades the performance of downstream tasks.
>
> 1. "Mitigating Impacts of Hyperedge Heterogeneity on Semi-Supervised Hypergraph Contrastive Learning" (K Yin et al., 2025)
> K. Yin et al. highlight that hypergraph neural networks suffer from two major issues: (1) During message passing, heterogeneous hyperedges mix information from neighbors belonging to different classes, causing the aggregated signals at a node to be contaminated and leading to poor representation learning. (2) Hyperedge heterogeneity impairs the model's discriminative capability, leading to blurred classification boundaries.
>
> These studies indicate that noise and irrelevant nodes incorporated into hyperedges—the hyperedge pollution problem proposed in our paper—increase the heterogeneity of hypergraphs and affect their construction. When a hypergraph contains too many heterogeneous hyperedges, its structure can no longer faithfully capture the core relational patterns of the dataset, making it difficult for the model to learn discriminative node representations and thus to perform well on subsequent node-level tasks. Therefore, the heterogeneity of hyperedges negatively impacts model performance on node-level tasks: higher heterogeneity correlates with poorer model performance.

---

> ### Author Response · Authors · 2025-11-16
> **Answer for Q2**
>
> 2.1 Thanks for your insightful comments.
>
> To address this, we profiled the time cost of each component in HPHNN. Specifically, we divided the training process into five modules: clustering, HP score computation, node and hyperedge attention coefficient calculation, loss calculation, and GNN initial embedding learning. Details are presented in the table below. The results indicate that clustering is the primary cause of computational inefficiency, accounting for 68.02% of the average runtime. This is because each hypergraph update step requires re-running K-means clustering, which substantially increases the overall computational cost of HPHNN.
> | Operation | Time (ms) | Percentage |
> |-----------|-----------|------------|
> | Total Time | 350.77 | 100% |
> | HP Score | 56.92 | 16.2% |
> |Loss Computation|14.78|4.2% |
> | KMeans Clustering | 238.61 | 68.0% |
> | Node-to-edge Attention | 3.22 | 0.9% |
> | Edge-to-node Attention | 31.10 | 8.9% |
> | GCN Initial Embedding | 6.14 | 1.8% |
>
> To improve the model's scalability, we revisited the clustering component in HPHNN. Specifically, we perform clustering only once during the initial training to obtain the initial hypergraph, which is then saved locally. Subsequent training iterations do not involve clustering operations but instead continuously optimize and refine the existing results. This approach significantly reduces computational time and GPU memory consumption. Additionally, we introduced a new hyperedge update strategy. We calculated direct similarity between nodes within hyperedges and measured hyperedge center similarity, designing a hyperedge splitting and merging strategy. While significantly boosting computational efficiency, the updated algorithm delivers identical performance to HPHNN on node-level tasks. Below are runtime and clustering accuracy metrics for the updated algorithm on classic datasets PubMed, Computers, and Products, where HPHNN_new denotes the updated approach. Results demonstrate that the updated algorithm achieves the lowest or second-lowest time complexity while substantially outperforming other baseline algorithms in accuracy, confirming its scalability.
> | Method | PubMed (Time/ACC) | Computers (Time/ACC) | Products (Time/ACC) |
> |--------|-------------------|----------------------|---------------------|
> | HGNN | 58.4ms / 79.39% | 52.3ms / 72.45% | 12418.6ms / 78.70% |
> | DHGNN | 517.6ms / 79.52% | 461.3ms / 73.59% | 115684.6ms / 80.46% |
> | TDHNN | 413.5ms / 87.48% | 306.4ms / 92.60% | 92465.7ms / 82.97% |
> | DHHNN | 383.0ms / 89.47% | 237.7ms / 92.01% | 78256.7ms / 84.29% |
> | HPHNN | 1837.5ms / 91.21% | 1658.4ms / 96.82% | 186729.4ms / 85.69% |
> | HPHNN_new | 223.9ms / 91.33% | 222.3ms / 95.24% | 28382.3ms / 85.42% |
>
> 2.2
> Thank you very much for your suggestions. Following your suggestion, we scaled the model parameters of HPHNN, reducing the dimension of the hidden layers, the number of convolutional layers, the Dropout rate, and the feature transformation complexity. As a result, the total number of trainable parameters is reduced to 6,248,647, which is smaller than that of all dynamic hypergraph baselines we compare with. Despite this substantial reduction in model size, our method still outperforms the existing baselines on node-level tasks overall. This indicates that HPHNN can achieve a favorable balance between parameter efficiency and predictive performance, offering a reasonable performance–cost trade-off.
> | Model | Parameters | Cora | Citeseer | Pubmed |
> |-------|------------|------|----------|--------|
> | DHGNN | 16,175,948 | 79.52% | 73.59% | 84.24% |
> | TDHNN | 6,466,055 | 88.67% | 79.57% | 87.48% |
> | DHHNN | 7,203,856 | 89.29% | 80.92% | 89.47% |
> | HPHNN (original) | 7,991,303 | 92.85% | 84.55% | 91.21% |
> | HPHNN (reduced) | 6,248,647 | 90.31% | 81.02% | 89.68% |
>
>
> We have incorporated all corresponding efficiency and scalability experiments into Appendix F.

---

> ### Author Response · Authors · 2025-11-16
> **Answer for Q3**
>
> 3.1
> Thank you for your valuable suggestions.
> Regarding the $\mathrm{Satur}(\mathcal{E})$ and $\\mathrm{Silh}(\widehat{\mathbf{Z}},\mathbf{Y})$ indicators mentioned in Eq. 12, there are indeed some gaps in the explanation. We have now added the missing definitions and citations for these two terms at Equation 12 on Page 6, and included the corresponding ablation study in Appendix F. Below is a more detailed analysis and interpretation.
> \begin{equation}\mathcal{S}_{\mathcal{H}}=\mathrm{Satur}(\mathcal{E})+\mathrm{Silh}(\widehat{\mathbf{Z}},\mathbf{Y})-\mathcal{H}\mathcal{P}(\mathcal{E})\end{equation}
>
> $\mathrm{Satur}(\mathcal{E})$ is a hyperedge saturation metric which was first proposed and employed by TDHNN (Totally Dynamic Hypergraph Neural Networks, Zhou et al., IJCAI 2023). Within the TDHNN framework, hyperedge saturation is defined as follows,
> \begin{equation}\mathrm{Satur}(\mathcal{E})=1-\frac{|E_{\mathrm{empty}}|}{|E|},\end{equation}
>
> where $E_{\mathrm{empty}}$ denotes an empty hyperedge with no nodes assigned. The purpose of introducing $\mathrm{Satur}(\mathcal{E})$ is to constrain undesirable phenomena such as hyperedge redundancy and excessive empty hyperedges during hyperedge reconstruction, thereby ensuring more stable and controllable structural updates.
>
> 3.2
> Thanks for your insightful comments.
> Regarding the definition and origin of $\\mathrm{Silh}(\widehat{\mathbf{Z}},\mathbf{Y})$ (Silhouette Score), the reviewer's observation is entirely accurate: $\\mathrm{Silh}(\widehat{\mathbf{Z}},\mathbf{Y})$ derives from the classic clustering evaluation metric silhouette score (Rousseeuw, 1987). Its definition is as follows:
> \begin{equation}\mathrm{Silh}(v)=\frac{b(v)-a(v)}{\max(a(v),b(v))},\end{equation}
> where a(v) is the average similarity (intra-class distance) of node v within its superedge and b(v) is the minimum average similarity (inter-class distance) from node v to other hyperedges. We employ this metric because hyperedge reconstruction is fundamentally a “cluster refinement” process. The Silhouette score effectively quantifies whether hyperedges are sufficiently compact internally and sufficiently separable between each other. Thus, it serves as a node-level structural quality constraint, ensuring that reconstructed hyperedges exhibit improved semantic consistency.
>
> To further validate the effectiveness of each component of Eq. 12, we conducted an ablation study with three metrics on Cora, Citeseer, and Pubmed datasets.
> | Model Configuration | Cora | Citeseer | Pubmed |
> |---------------------|------|----------|--------|
> | Full Model | 92.85% | 84.55% | 91.21% |
> | w/o Satur | 90.12% | 81.73% | 88.47% |
> | w/o Silh | 90.56% | 82.15% | 88.89% |
> | w/o HP | 89.34% | 80.92% | 87.68% |
>
> The experimental results in the above table clearly demonstrate the effectiveness of each component in Eq. 12. Across the Cora, Citeseer, and Pubmed datasets, the proposed model achieves the highest performance when all the metrics incorporated in the model formulation are fully retained. Removing key components, particularly the HP score, results in substantial performance degradation. This indicates that all metrics play crucial and complementary roles in enhancing the capability of the model to capture meaningful features from the hypergraph, and their integration is essential for achieving the optimal predictive performance.

---

### Note · Program_Chairs · 2026-01-17
**Submission Desk Rejected by Program Chairs**

The following references in this submission do not refer to real documents and/or have major errors in bibliographic information:

 Matthew Topping, Sebastian Ruder, and Chris Dyer. Understanding over-smoothing in graph neural networks. In Proceedings of the 39th International Conference on Machine Learning (ICML), 2022.
Wei Jin, Tengfei Ma, Qian Gu, Siwei Han, Jiliang Tang, and Qiaozhu Chen. Pro-gnn: Robust graph neural network against noisy and adversarial attacks. In Proceedings of the 37th International Conference on Machine Learning (ICML), 2020.
Peng Sun, Wen Jin, Zengfeng Wang, and Ying Yang. Idgl: Independent and identically distributed graph learning for robust graph neural networks. In Proceedings of the 26th ACM SIGKDD International Conference on Knowledge Discovery Data Mining (KDD), pp. 978-986, 2020.